# Quantification of Physiological Parameters of Rice Varieties Based on Multi-Spectral Remote Sensing and Machine Learning Models

**Shiyuan Liu [1,†], Bin Zhang [2,†], Weiguang Yang [3,†], Tingting Chen [1], Hui Zhang [1], Yongda Lin [3], Jiangtao Tan [1], Xi Li [1], Yu Gao [1], Suzhe Yao [1], Yubin Lan [3] and Lei Zhang [1,*]**

[1] College of Agriculture, South China Agricultural University, Guangzhou 510642, China
[2] Guangdong Key Laboratory of New Technology for Rice Breeding,
  Rice Research Institute of Guangdong Academy of Agricultural Science, Guangzhou 510640, China
[3] College of Electronic Engineering (College of Artificial Intelligence), South China Agricultural University,
  Guangzhou 510642, China
* Correspondence: zhanglei@scau.edu.cn
† These authors contributed equally to this work.

**Abstract:** Estimating plant physiological indicators with remote sensing technology is critical for ensuring precise field management. Compared with other remote sensing platforms, low-altitude unmanned aerial vehicles (UAVs) produce images with high spatial resolution that can be used to clearly identify vegetation. However, the information of UAV image data is relatively complex and difficult to analyze, which is the main problem limiting its large-scale use at present. In order to monitor plant physiological indexes from the multi-spectral data, a new method based on machine learning is studied in this paper. Using UAV for deriving the absorption coefficients of plant canopies and whole leaf area, this paper quantifies the effects of plant physiological indicators such as the soil and plant analyzer development (SPAD) value, whole leaf area, and dry matter accumulation on the relationship between the reflectance spectra. Nine vegetation indexes were then extracted as the sensitive vegetation indexes of the rice physiological indicators. Using the SVM model to predict the SPAD value of the plant, the mean squared error (MSE), root mean squared error (RMSE), mean absolute error (MAE), mean absolute percentage error (MAPE), and symmetric mean absolute percentage error (SMAPE) values of the model were 1.90, 1.38, 0.13, 0.86, and 4.13, respectively. The results demonstrate that the rice plants display a considerable biochemical and spectral correlation. Using SVM to predict the SPAD value has a better effect because of a better adaptation and a higher accuracy than other models. This study suggests that the multi-spectral data acquired using UAV can quickly estimate field physiological indicators, which has potential in the pre-visual detection of SPAD value information in the field. At the same time, it can also be extended to the detection and inversion of other key variables of crops.

**Keywords:** rice; physiological parameters; UAV; machine learning

## 1. Introduction

Rice serves as the staple food for half of the world's population [1]. Rice physiological parameters, including chlorophyll content, whole leaf area, and dry matter accumulation, are considered as the most important plant indexes, which can effectively reflect plant growth [2,3]. As the key photosynthetic pigment, chlorophyll is responsible for the harvesting of solar radiation and the conversion of chemical energy, and it is directly involved in photosynthetic potential and primary production [4]. Moreover, chlorophyll content is well-linked with nitrogen concentration, which is a key parameter in promoting the growth of plants and evaluating vegetation nutritional conditions. Aboveground biomass and whole leaf area are important agronomic parameters reflecting crop growth [5], and

there is a close relationship between them [6]. At different stages of crop growth and development, the spectral characteristics of crop canopy in the image are also different due to the changes in crop canopy structure, leaf morphology, and physiological and ecological characteristics [7]. As the main place of photosynthesis, the whole leaf area of the plant has a direct impact on the physiological state of the plant [8]. The dry matter accumulation of plants directly reflects the accumulation of organic matter in crops. From the perspective of agriculture, chlorophyll, whole leaf area, and dry matter could be used as indicators to quantify crop photosynthetic capacity and plant productivity, or to estimate plant growth status [9,10]. Therefore, the real-time and accurate acquisition and the temporal and spatial distribution of the crop chlorophyll, whole leaf area, and dry matter are of significance for modern agricultural production.

At present, the main methods for determining the content of chlorophyll are ultraviolet spectrophotometry, fluorescence analysis, and chlorophyll analysis in vivo. SPAD 502 is an effective tool for measuring plant chlorophyll. The 502 reading is highly linear, with the chlorophyll value being measured using chemical analysis [11,12]. Many studies have shown that the results of chemical experiments that are used to measure chlorophyll content are almost the same as those that are measured using SPAD 502, indicating that SPAD can replace chlorophyll content [13]. The detection of whole leaf area and dry accumulation is carried out manually. These methods are inefficient in actual agricultural production. Therefore, a more effective method is crucial for the accurate detection of rice physiological parameters [14,15].

Remote sensing is an effective technology for obtaining ground information based on the principle of electromagnetic radiation, which has been proven to be an effective and non-destructive technology for detecting physiological parameters at different spatial scales [10–12]. During vegetation growth, one way to monitor the physiological parameters of the plant canopy is to retrieve the physiological characteristics of the plant canopy through the reflection characteristics of the plant canopy [16,17]. Between 400 nm and 1000 nm, the vegetation shows typical reflection characteristics. The absorptions of pigments (mainly 430/660 nm—chlorophyll a and 450/640 nm—chlorophyll b, and 450 nm—other pigments, such as carotenoids and lutein) are controlled within the visible wavelength range (400–700 nm), and the near-infrared range (700–1100 nm) is controlled through the reflection process within the leaf surface layer [18,19]. At the same time, the electromagnetic energy reflection in the near-infrared spectrum (NIRS) range of 800–2500 nm has also been studied for the non-destructive measurement of organic materials such as food, agricultural products, and forest products. NIR absorption is mainly attributed to the overtones and combinations of vibration bands involving C-H, O-H, and N-H in the infrared (IR) region. Compared with the infrared region, the absorption of NIR energy is weak, leading to the nondestructive measurement of high-density and concentrated organic materials. Since the molar absorptivity of water in the near-infrared range is 1/1000–1/10,000, it is also useful for samples with a high water content, such as fruits [20]. Based on the above point of view, according to the electromagnetic characteristics of substances, the target content or composition can be determined using remote sensing without using chemical substances.

At present, the research objectives of the forecasters focus on estimating crop planting area and estimating the overall growth through satellite and other large-scale remote sensing data, but there is a lack of accurate prediction models for small areas. Most of the selected models are simple linear models [14]. The result may be changed in the external environment and through human operations. In recent years, with the development of electronics technology, UAV (unmanned aerial vehicle) remote sensing has occupied an increasingly important position in agricultural monitoring by virtue of its practicality, flexibility, and highly temporal nature [21]. The diagnosis of crop nutritional status has become an UAV remote sensing technology application development trend. The methods, based on visible light photos, could calculate multiple vegetation indices, and they use regression algorithms to build FVC models that are suitable for different growing seasons,

growth stages, and crop water stress [22]. However, visible light photographs contain little information, and it is difficult to comprehensively reflect the physiological situation of the plants. Unlike visible light photos, multi-spectral data can describe various characteristics that are associated with the biochemical and physiological traits of the targets. Yang et al. used 12-band cameras to obtain multi-time images, and selected the main typical frequency bands that constitute the green, red, red edge, and near-infrared VI to analyze their spectra throughout the growing season and texture changes [23]. After the mathematical combination of the plot-level spectrum and texture values, a new index was constructed to estimate the VI index corresponding to rice LAI. Liu [24] estimated the whole leaf area and the dry biomass by modeling different vegetation indices to obtain the correlation magnitude and accuracy of each index. Carmona [25] proposed the calculation of the Normalized Vegetation Index (NAVI) for the estimation of chlorophyll content from remote sensing data using multi-spectral images acquired using the Proba/CHRIS sensor. Zhao [26] carried out experiments on cotton and calculated the reflectance ratio vegetation index (RVI), normalized difference vegetation index (NDVI), enhanced vegetation index (EVI), wide dynamic range vegetation index (WDRVI), and several hyperspectral reflectance indexes. The previous studies show that it is feasible to use the vegetation index to predict plant physiological parameters.

The successful application of remote sensing products depends on the data characteristics and the types of analytics applied. Machine learning is the cornerstone of artificial intelligence, and it is an interdisciplinary subject involving probability theory, statistics, linear algebra, advanced mathematics, algorithm complexity theory, and other fields [27,28]. Relying on the powerful performance of modern computers, machine learning algorithms can simulate human learning behavior, mine useful rules and knowledge from a large amount of data information, and constantly reorganize the knowledge structure to achieve the goal of multiple iterations of self-improvement [29]. Therefore, compared with traditional algorithms, the main advantage of machine learning algorithms is that they can significantly improve upon the simulation accuracies of phenomena and processes [30]. Among them, random forest (RF), support vector regression (SVR), and other nonparametric models have recently become more and more popular in analyzing complex remote sensing data [31]. RF is a tree-based supervised learning algorithm that is used to solve classification and regression problems. Since the RF model is composed of many decision trees, it can generate more accurate and stable predictions [32,33]. SVM is a supervised learning method for retrieving biophysical parameters from remote sensing data. The selection of the kernel function type is a key aspect of the applicability of the SVM model. Because the problem of dimension disaster and nonlinear separability is overcome by using the kernel function method, the complexity of calculation is not increased when mapping to a high-dimensional space [34].

Nitrogen (N) plays a vital role in crop growth, development, and yield formation. The amount of chemical N fertilizer has increased dramatically from 11.3 Tg/year to 107.6 Tg/year over the past four decades [3]. China's rice production accounts for 20% of the world's total, but consumes 35% of the world's total N fertilizer applied across the globe [35]. Most of the applied N is lost to the environment, with only 30–50% being used by crops [36].

In this paper, UAV is used to quickly obtain the reflectivity of plant tube layer and to calculate the vegetation index. This paper creatively uses multiple vegetation indexes to jointly predict plant physiological indicators, and uses the idea of machine learning to achieve the lightweight and non-destructive monitoring of plant parameters. Compared with single index prediction, the multi-vegetation index has achieved better results for the prediction of rice physiological parameters. Through the rapid prediction of this rice physiological index, we can obtain the plant growth status more stereoscopically and provide a reference for breeders. Therefore, the overall goal of this research is to develop a methodology through the detection of the physiological parameters of different rice varieties over different periods, using rice canopy multi-spectral images. Thus, the



specific objectives were to (1) study the effect of nitrogen application rate on different rice varieties and different growth stages, and (2) to use multi-spectral images to predict rice physiological parameters.

## 2. Materials and Methods

### 2.1. Experiment Site

Ground-based field experiments were conducted at Baiyun Experimental base, Guangzhou Academy of Agricultural Sciences, Guangzhou City of Guangdong Province (longitude and latitude: 113.44, 23.39; altitude: 18 m) during the 2020 and 2021 cropping season. The region has a typical tropical monsoon climate zone, with a mild climate and significant oceanic climate characteristics. The location of the study area for ground-based field observations is shown in Figure 1.

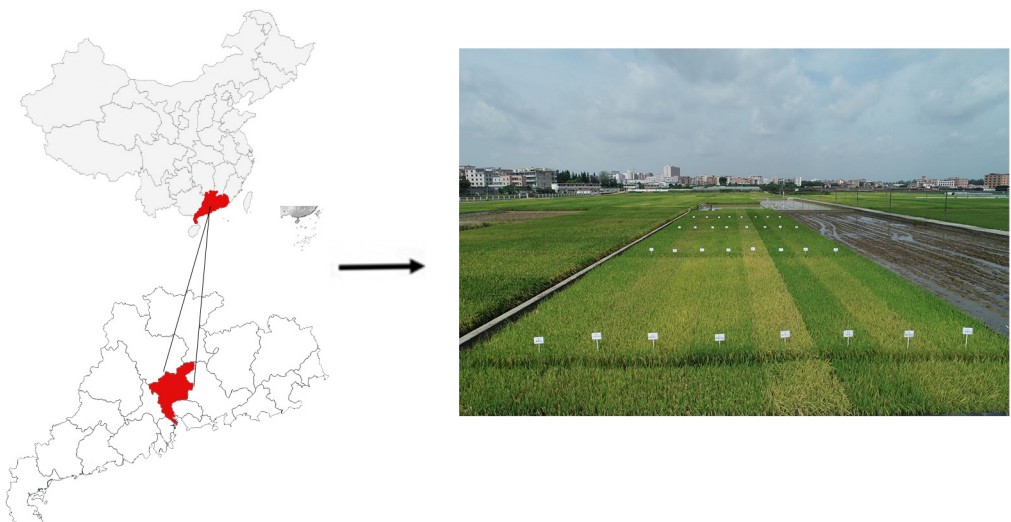

Longitude: 113.44°, Latitude:23.39°, Altitude: 18m.

**Figure 1.** Location of the experimental farm for ground-based and UAV-based field observations.

### 2.2. Treatment and Experimental Design

The field experiment was laid out using a split-plot arrangement based on a randomized complete block design with four N application rates (0, 5, 7, and 9 kg/666 m$^2$, abbreviated as N0, N1, N2, and N3, respectively) and eight different rice varieties (Mei Zhanxiang 2, Xiang Yaxing, 19 Xiang, Ruanhua, Qing Xiangyou 033, Nan Jingxiang, Er Guangxiang 3, and Li Xiangzhan, abbreviated as H1 to H8, respectively). The gross size of each plot was 36 m$^2$, and the row spacing of the transplanted plants was 6 × 5 inches, with 11 plants ×80 rows. Three main seedlings were inserted into each family. The outermost row on both sides of each plot was considered as a border and was not used for data collection, to avoid border effects. An irrigation and seedling drainage ditch 0.6 m wide and a ridge 0.3 m wide were set up, surrounded by protection lines. The communities were separated by ridges. The ridges were covered with film, with independent drainage and irrigation (Figure 2). Plot irrigation was used with unpolluted water. N fertilizer (urea) was split-applied artificially at basal, tillering, and heading dressings, and the N-splitting patterns were 50%, 30%, and 20%, respectively. The base fertilizer was applied 1–2 days before transplantation, and the tiller fertilizer was applied 5 days after transplantation. The basal fertilizers, including phosphorus (4 kg/666 m$^2$) as calcium superphosphate and potassium (8 kg/666 m$^2$) as potassium chloride, were broadcast and incorporated into the topsoil via rotary tillage from one day before sowing. The management of water slurry with the prevention and control of diseases, pests, and weeds was carried out in accordance with high-yield fields.

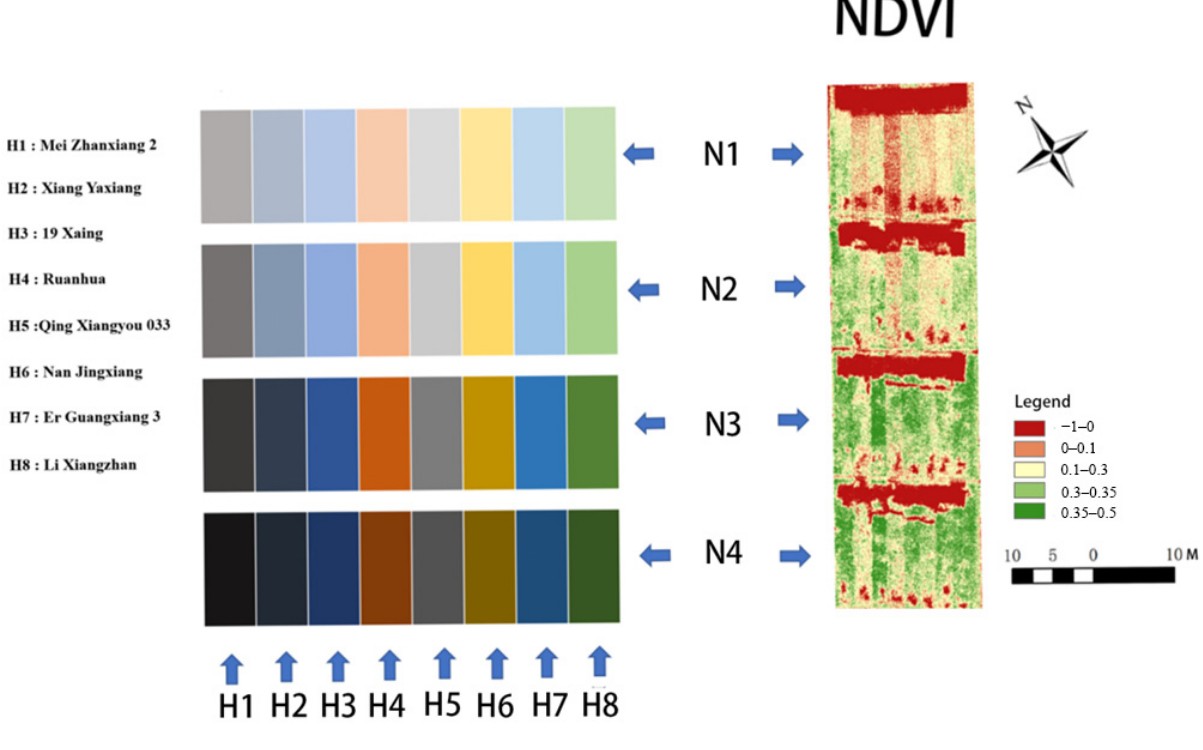

**Figure 2.** Experimental farm district distribution maps.

### 2.3. Ground-Based Field Observations

The growth period of earth rice in this region is from early March to July. We chose two important philological phases, namely the heading stage (7 May 2021) and the maturity stage (7 June 2021), for field data measurements. In this work, three rice parameters were determined, including the whole leaf area, dry matter accumulation, and SPAD. For an accurate determination of SPAD, we used a portable SPAD meter (SPAD value-502 Plus; Minolta Camera Co., Osaka, Japan) to randomly select 15 normal-growing rice plants for each experimental plot, and to determine the SPAD value of 3 leaves. The preliminary processing of the data was performed to calculate the average value of each observation sample, and the average value was recorded as the test area SPAD value, respectively. Finally, 64 average values were obtained as cell statistics, and 64 groups of whole leaf area and dry weight were obtained through the same method (Table 1).

**Table 1.** Summary of physiological parameters of rice.

| Type | Number of Samples | Max | Min | Average | Standard Deviation |
|:---:|:---:|:---:|:---:|:---:|:---:|
| SPAD value | 64 | 41.85 | 29.52 | 35.54 | 2.61 |
| Dry matter accumulation (kg/666 m$^2$) | 64 | 828.61 | 209.94 | 467.24 | 210.86 |
| Whole leaf area (cm$^2$) | 64 | 2513.47 | 981.31 | 1633.36 | 428.17 |

### 2.4. The UAV Model

The camera equipment used in this article was the DJI Phantom 4 multi-spectral drone, using 6 1/2.9″ CMOS, including an RGB sensor for visible light imaging and five monochromatic sensors for multi-spectral imaging. The take-off accumulation was 1487 g and the maximum flight time was about 27 min. The hover accuracy range when RTK was enabled and working normally, and the vertical accuracy was ± 0.1 m. The horizontal accuracy was ± 0.1 m. The UAV automatically flies through the GS PRO planning flight

mission, and the integrated spectral sunlight sensor on the top can capture solar radiation, thereby maximizing the accuracy and the consistency of data collection at different times of the day. When combined with the processed data, this information helps to obtain more accurate remote sensing results. Finally, the remote sensing data are spliced through the Pix4d software to obtain red, blue, green, near-infrared, and edge bands. The spectral sensor information is shown in Table 2.

**Table 2.** Spectral sensor information.

| Band | Central Wavelength (nm) | Width (nm) |
|---|---|---|
| Edge | 730 | 32 |
| Near-infrared | 840 | 52 |
| Green | 560 | 32 |
| Red | 650 | 32 |
| Blue | 450 | 32 |

### 2.5. UAV-Based Field Observations

The UAV-based experimental farm is located at the Baiyun Experimental Base of Guangzhou Academy of Agricultural Sciences in Guangzhou City of Guangdong Province, Chain. This area has a typical tropical monsoon climate. Combining the five bands of the stitched multi-spectral image for different operations can yield different vegetation indices. The nine vegetations, including RVI, NDVI, EVI, GNDVI, NLI, SAVI, OSAVI, LCI, and SIPI2, are closely related to the growth of plants. The calculation formula was shown in Table 3. The physiological parameters of plants are recorded in a certain area. In order to establish a physiological model more accurately, it is necessary to calculate the average value of each area with similar physiological conditions as the index of the area. The original image was pre-processed using Pix4dmapper and ENVI software, including irradiance calibration, mixed noise filter (MNF) denoising, geometric correction, and image mosaicking. The system, combined with an accurate inertial measurement unit and precise gimbals, provided stable and high-quality spatial multi-spectral imagery.

**Table 3.** Vegetation index and calculation formula.

| Vegetation Index | Formula | Reference |
|---|---|---|
| RVI | Rnir/R | Jordan (1969) [37] |
| NDVI | (RNIR-R)/(RNIR + R) | Tucker (1979) [38] |
| EVI | $2.5 \times$(NIR-R)/(NIR + 6 R − 7.5 B + 1) | Hui et al. (1995) [39] |
| GNDVI | (NIR-G)/(NIR + G) | Anatoly et al. (1996) [40] |
| NLI | (NIR × NIR-R)/(NIR× NIR + R) | Goel et al. (1994) [41] |
| SAVI | $(1 + 0.5) \times$ (NIR-R)/(NIR + R + 0.5) | Huete (1988) [42] |
| OSAVI | $(1 + 0.16) \times$ (NIR-R)/(NIR + R + 0.16) | Geneviève et al. (1996) [43] |
| LCI | (NIR-RedEdge)/(NIR + RedEdge) | Su et al. (2005) [44] |
| SIPI2 | (NIR-Green)/(NIR − Red) | Yue et al. (2018) [45] |

### 2.6. Machine Learning Modeling

The support vector machine (SVM) informs an excess glider from the input field that disintegrates a particular training dataset and permits distance on both sides of the hyperplane from the nearest instances. For the optimization process, the model generates a penalty factor for the generated error classification, punishes the error classification through the penalty factor, and obtains the total penalty by adding the penalty.This technique has a hyperplane that is responsible for the minimization of the sum of the reciprocal value of the margin and the total penalty, which is obtained by adding the penalties. Random forest algorithm is an algorithm that builds multiple trees to train the model and predict the outcome. This algorithm has the advantages of processing large amounts of data, and a high accuracy of results, balancing errors, and speed. In multiple linear regression (MLR), several inputs are connected through a single regression equation for a given output. In

recent years, these techniques were applied successfully over many fields to obtain the desired results. In this study, a total of 64 sets of data on the vegetation index and plant physiological indicators of each sample were used as the training data, and the data were allocated according to a ratio of 2 to 1 between the training set and the test set to continue the training of the SVM, random forest model, random tree model, and linear regression model. The model was trained using Anaconda software based on pandas and sklearn.

In order to judge the goodness of fit between the regression models and the original data, the coefficient of determination ($R^2$) was chosen as the criterion, and the closer the $R^2$ was to 1, the better the fit. At the same time, the parameters for determining the accuracy of the regression model were root mean square error (RMSE), average absolute error (MAE), symmetric average absolute percentage error, average trivial error (MSE), and average absolute percentage error (MAPE). The smaller the values of RMSE, MAE, MSE, and MAPE were, the more accurate the model. RMSE, MAE, MSE, and MAPE were expressed through the following formula [14]. The technical route is shown in Figure 3.

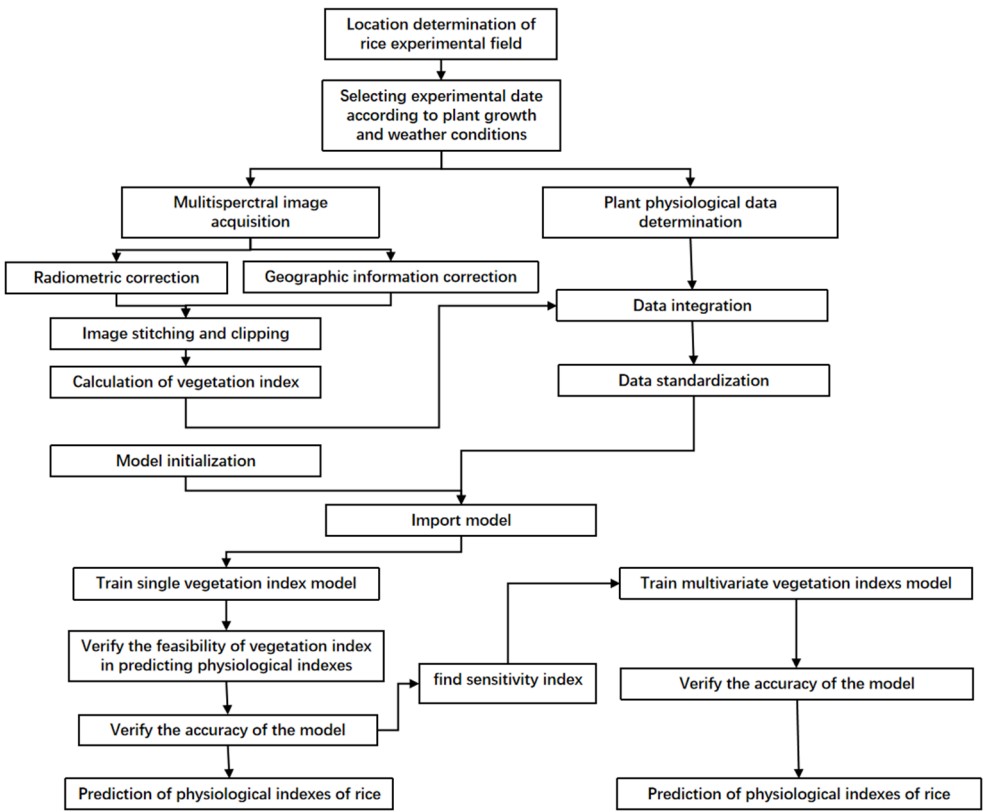

**Figure 3.** Framework of multispectral mode.

## 3. Results

### 3.1. Rice Biochemical Variation

Comparing the changes in the same rice variety under different nitrogen application conditions, we can see that there are differences in the biochemical parameters between the tillering stage and the full heading stage. At the tillering stage, the SPAD content of most plant leaves changes significantly under the conditions of N1 and N2, which proves that appropriate nitrogen application is conducive to the synthesis of chlorophyll in plant leaves, but with a further increase in nitrogen application, the SPAD in leaves does not continue to change significantly. The dry matter weight and whole leaf area of rice changed significantly with the increase in nitrogen application. However, at the full heading stage, the SPAD of most plants did not change significantly with the change in nitrogen application, and the dry matter weight and whole leaf area of some plants changed significantly under N1 and N2 treatment. However, with the further increase in nitrogen application, the dry matter

weight and whole leaf area did not continue to change significantly. Comparing the changes in different rice plants under the same nitrogen application conditions, it was found that different plants had different needs and sensitivities to nitrogen. Under the N1 treatment with low nitrogen application, the SPAD value of ivory Xiangzhan and Guangxiangzhan 3 was significantly higher than those of other plants, but with the increase in nitrogen application, the SPAD value of ivory Xiangzhan decreased compared with other plants. Qingxiangyou 033 had the largest horizontal plant dry matter weight under N2 treatment with low nitrogen application, but with the increase in nitrogen application, the plant dry matter weight of Ruanhuayou Jinsi increased rapidly, which was significantly higher than Qingxiangyou 033 under N4 treatment (Figure 4). Detailed tabular data are presented in Appendix A—Table A1.

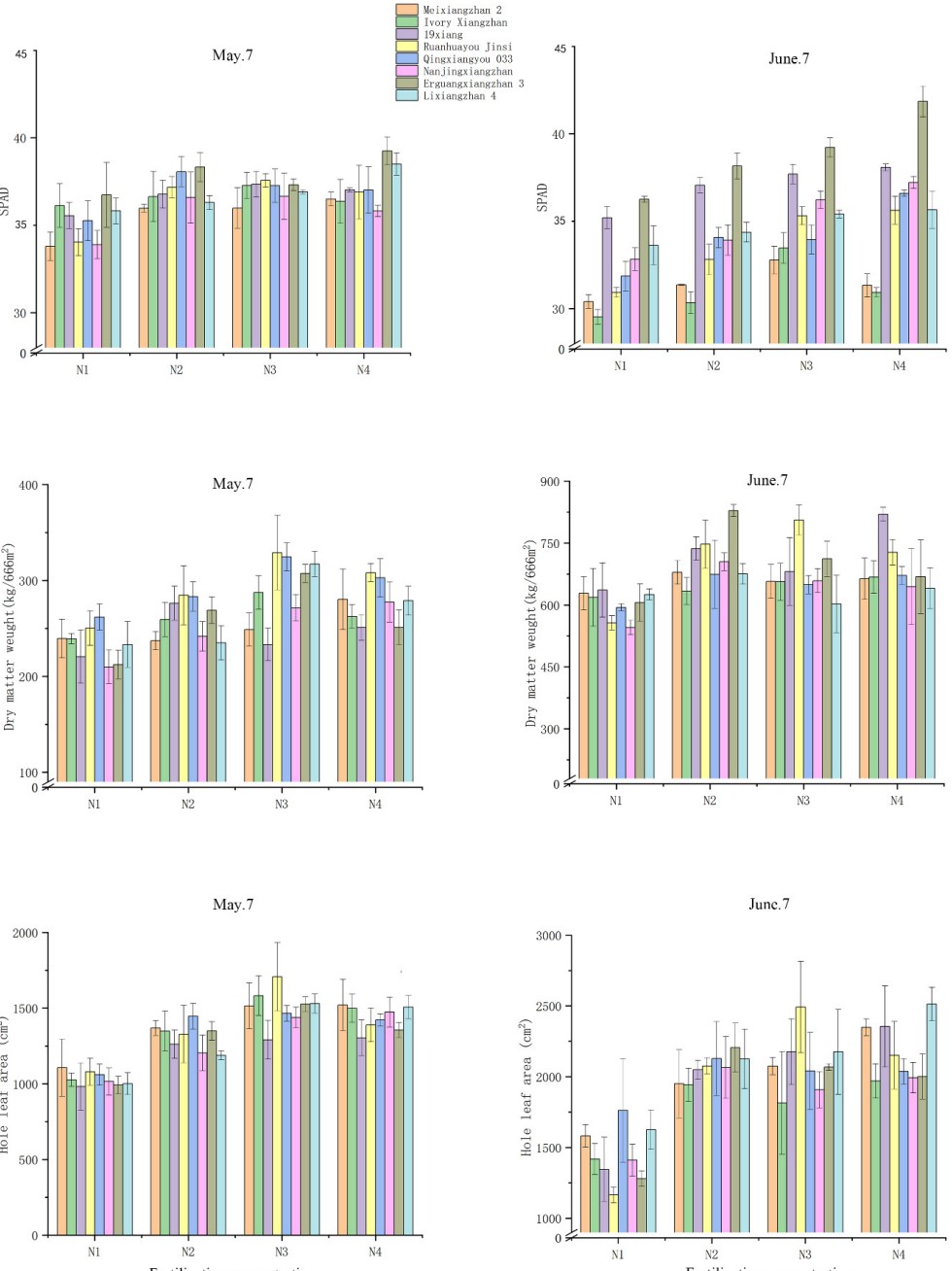

**Figure 4.** Physiological parameters of peanut under different nitrogen application levels and growth stages.

### 3.2. Change of Rice Vegetation Index

By comparing the different vegetation indexes in Figure 5, it can be found that most of the indexes increase with an increase in the nitrogen application in the plot. There are obvious changes among N1, N2, and N3, but N3 and N4 do not change significantly. Different plants also have great differences. Ruanhuayou Jinsi has a low vegetation index under N0, but this increases rapidly with the increase in nitrogen application. Compared with other plants, Guangxiangzhang 3 changed slowly with nitrogen application (Figure 5).

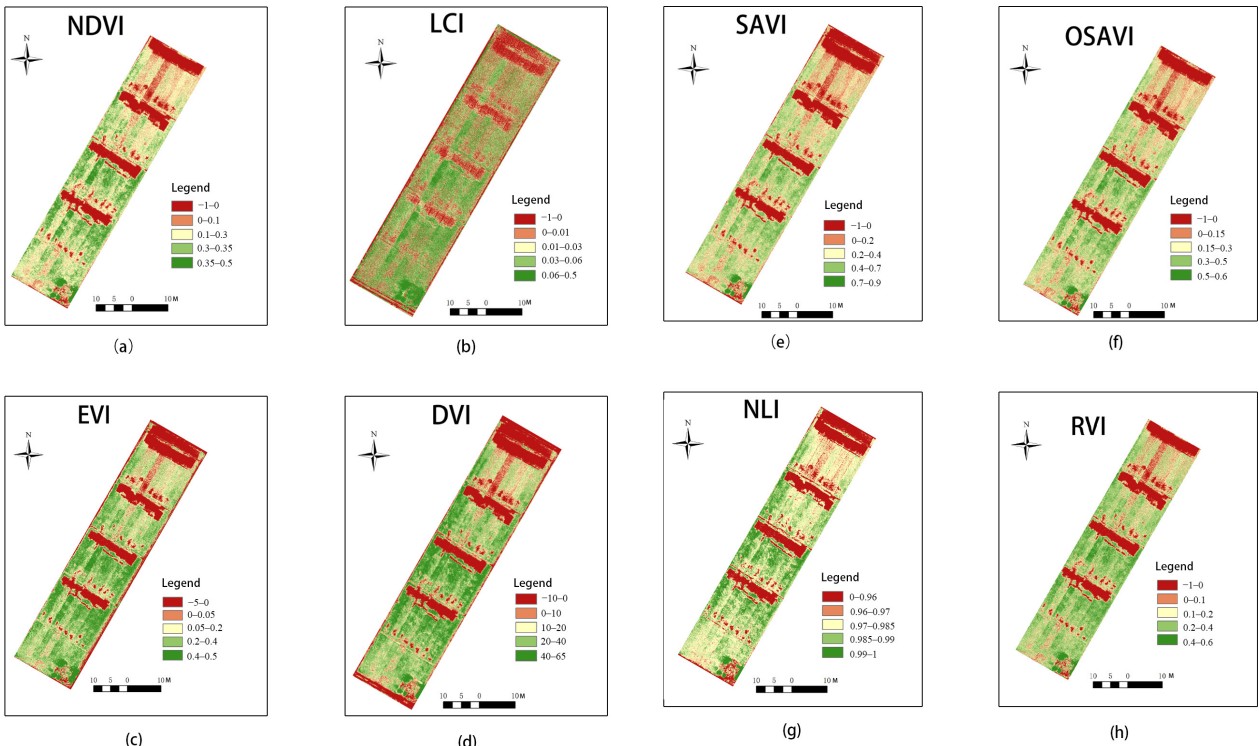

**Figure 5.** (**a–h**) shows the canopy vegetation index of peanut at full heading stage.

### 3.3. Single-Factor Regression Modeling

$R^2$ is the ratio of the sum of squares of regression and the sum of squares of total deviation. As a standard for evaluating the goodness of fit of a model, it is called the sample determination coefficient. The closer its value is to 1, the higher the goodness of fit of the model. As is shown in Table 4, the high correlation coefficient $R^2$ of some models indicates that the remote sensing data and the physiological data are correlated. Compared with other models, the appearance of the tree model, the forest model, and the SVM models are better than the linear model. Compared with other remote sensing parameters, EVI, LCI, and OSAVI show better results; specifically, the fitting coefficients of dry matter accumulation and whole leaf area are higher than 0.75 in the tree model and in the forest model. However, the overall performance of SIPI2 does not meet the expectation, and it is difficult to predict the physiological values.

### 3.4. Analysis of Single-Factor Regression Model Accuracy

As shown in Table 5, the overall MAPE (mean absolute percentage error) performance of the SVM model and the linear regression model of the four models are better than for the other models. Among the nine remote sensing parameters, LCI, NDVI, and SAVI performed the best, and SIPI2 and OSAVI performed the worst. The prediction results of the three physiological parameters and the prediction results of the SPAD value are relatively accurate, but the prediction results of dry accumulation and whole leaf area have a large deviation. By comparing all of the models, it was found that the linear model and the SVM regression model has the best prediction effects of SPAV via LCI, NDVI, and SAVI.

The model MAPE reached 2.98, 3.55, 3.21, 6.64, 6.17, and 6.15. The complete single-factor regression model accuracy document is shown in Appendix B—Table A2.

**Table 4.** Regression modeling of rice physiological parameters with vegetation index.

| Type ($R^2$) | Vegetation Index | Linear Regression Model | Random Tree Model | Random Forest Model | SVM |
|---|---|---|---|---|---|
| SPAD Value | EVI | 0.45 | 0.02 | 0.26 | 0.41 |
| | GNDVI | 0.33 | 0.33 | 0.07 | 0.29 |
| | LCI | 0.47 | 0.11 | 0.61 | 0.57 |
| | NDVI | 0.53 | −0.30 | 0.09 | 0.32 |
| | NLI | 0.21 | 0.15 | 0.04 | 0.31 |
| | OSAVI | 0.33 | 0.49 | 0.62 | 0.50 |
| | RVI | 0.26 | 0.15 | −0.09 | 0.23 |
| | SAVI | 0.40 | 0.18 | 0.23 | 0.26 |
| | SIPI2 | 0.26 | −0.13 | −0.24 | 0.19 |
| Dry matter accumulation | EVI | 0.83 | 0.92 | 0.95 | 0.11 |
| | GNDVI | 0.97 | 0.97 | 0.94 | 0.84 |
| | LCI | 0.73 | 0.46 | 0.69 | 0.36 |
| | NDVI | 0.89 | 0.96 | 0.92 | 0.92 |
| | NLI | 0.70 | 0.81 | 0.83 | 0.31 |
| | OSAVI | 0.82 | 0.80 | 0.96 | 0.87 |
| | RVI | 0.84 | 0.98 | 0.98 | 0.92 |
| | SAVI | 0.78 | 0.94 | 0.97 | 0.88 |
| | SIPI2 | 0.10 | 0.94 | 0.94 | 0.87 |
| Whole leaf area | EVI | 0.29 | 0.72 | 0.87 | 0.00 |
| | GNDVI | 0.70 | 0.38 | 0.36 | 0.02 |
| | LCI | 0.36 | 0.36 | 0.06 | 0.05 |
| | NDVI | 0.39 | 0.71 | 0.05 | 0.03 |
| | NLI | 0.05 | 0.05 | 0.36 | 0.02 |
| | OSAVI | 0.45 | 0.73 | 0.71 | 0.49 |
| | RVI | 0.36 | 0.40 | 0.41 | 0.02 |
| | SAVI | 0.59 | 0.45 | 0.83 | 0.83 |
| | SIPI2 | 0.23 | 0.40 | 0.56 | 0.34 |

**Table 5.** Comparison of prediction accuracy of single-factor regression modeling.

| Type (MAPE) | Vegetation Index | Linear Regression Model | Random Tree Model | Random Forest Model | SVM |
|---|---|---|---|---|---|
| SPAD | EVI | 4.30 | 6.53 | 7.06 | 6.02 |
| | GNDVI | 3.75 | 6.38 | 6.23 | 4.35 |
| | LCI | 2.98 | 6.55 | 7.16 | 6.64 |
| | NDVI | 3.55 | 5.60 | 5.82 | 6.17 |
| | NLI | 3.30 | 7.21 | 5.81 | 4.82 |
| | SAVI | 3.21 | 6.97 | 5.53 | 5.54 |
| | RVI | 3.82 | 6.26 | 7.92 | 5.04 |
| | SAVI | 3.94 | 7.82 | 6.02 | 6.15 |
| | SIPI2 | 5.20 | 6.37 | 5.64 | 5.34 |
| Dry matter accumulation | EVI | 16.38 | 43.42 | 56.49 | 68.53 |
| | GNDVI | 9.75 | 54.36 | 55.79 | 36.28 |
| | LCI | 23.31 | 62.17 | 62.20 | 49.32 |
| | NDVI | 11.33 | 58.03 | 58.59 | 72.96 |
| | NLI | 24.19 | 59.36 | 53.32 | 55.54 |
| | SAVI | 13.75 | 53.95 | 54.76 | 43.04 |
| | RVI | 17.20 | 61.59 | 56.17 | 90.88 |
| | SAVI | 19.92 | 53.53 | 60.65 | 53.52 |
| | SIPI2 | 49.22 | 60.62 | 60.11 | 46.40 |

**Table 5.** *Cont.*

| Type (MAPE) | Vegetation Index | Linear Regression Model | Random Tree Model | Random Forest Model | SVM |
|---|---|---|---|---|---|
| Whole leaf area | EVI | 19.77 | 26.36 | 30.23 | 22.66 |
| | GNDVI | 13.12 | 37.82 | 36.00 | 19.16 |
| | LCI | 13.52 | 27.81 | 23.02 | 16.47 |
| | NDVI | 12.97 | 24.94 | 31.77 | 18.66 |
| | NLI | 16.76 | 27.89 | 25.38 | 17.84 |
| | SAVI | 12.01 | 28.91 | 25.19 | 19.71 |
| | RVI | 14.90 | 30.68 | 32.76 | 20.04 |
| | SAVI | 10.71 | 21.95 | 28.99 | 24.60 |
| | SIPI2 | 20.97 | 28.69 | 22.24 | 21.94 |

*3.5. Analysis of Multi-Factor Regression Model Accuracy*

Considering the powerful data analysis ability of machine learning, and providing more abundant spectral data for the prediction of physiological indexes, we used all nine spectral indexes mentioned as inputs to build a regression model. The result of the model text is shown in Table 6. It can be seen from the table that the SVM and forest models perform the best among the other models, which are distinctly higher than the random tree and linear regression models. Especially, the prediction effect on the SPAD value is excellent. The MSE, RMSE, RMAE, MAPE, and SMPE of the prediction, which uses the SVM to predict the SPAD value, are 1.90, 1.38, 0.13, 0.86, and 4.13, respectively. At the same time, the random forest has also achieved adequate results. The MSE, RMSE, RMAE, MAPE, and SMAPE of the model are 2.22, 1.49, 0.55, 1.18, and 7.30. However, the prediction effect of dry accumulation and whole leaf area do not meet the expectation, and it is difficult to accurately predict them effectively.

**Table 6.** Comparison of prediction accuracy of multi-factor regression mode.

| Type | Model Type | RMSE | MAE | MAPE | SMAPE |
|---|---|---|---|---|---|
| SPAD value | Linear regression | 1.71 | 0.17 | 1.38 | 3.80 |
| | Random tree | 1.91 | 1.46 | 10.70 | 10.82 |
| | Random forest | 1.49 | 0.55 | 1.18 | 7.30 |
| | SVM | 1.38 | 0.13 | 0.86 | 4.13 |
| Dry matter accumulation | Linear model | 59.11 | 55.33 | 15.13 | 14.01 |
| | Random tree | 210.85 | 126.58 | 48.02 | 37.31 |
| | Random forest | 24.27 | 17.17 | 60.58 | 49.54 |
| | SVM | 201.25 | 182.80 | 61.96 | 44.79 |
| Whole leaf area | Linear regression | 363.39 | 315.38 | 25.46 | 21.76 |
| | Random tree | 146.26 | 108.63 | 24.80 | 23.96 |
| | Random forest | 201.89 | 161.34 | 27.07 | 26.76 |
| | SVM | 371.36 | 326.82 | 24.68 | 22.72 |

**4. Discussion**

The physiological processes of plants, chlorophyll concentration, dry matter accumulation, and whole leaf area determine the state of plants. At the same time, nitrogen is one of the most important nutritional indicators of crop growth. The amount of nitrogen application will greatly affect the changes in chlorophyll concentration, dry matter accumulation, and whole leaf area. Shi conducted different nitrogen application treatments on wheat and measured the chlorophyll contents and whole leaf areas of plant leaves. The results showed that the chlorophyll concentration and whole leaf area increased first, and then decreased with the change in nitrogen application [46]. Liao measured the dry matter mass of winter melon under different nitrogen treatments, and finally concluded that the dry matter mass of winter melon increased first and then decreased with the increase in

nitrogen application [47]. This is different from the results of our experiment. It should be considered that the maximum nitrogen treatment set in the experiment did not inhibit the plant. At the same time, comparing different vegetation indexes, it can be found that most indexes increase with the increase in nitrogen application in the plot, which is the same as the change trend of chlorophyll concentration, dry accumulation, and whole leaf area, which mirrors the research results of Ni. He obtained the same trend by analyzing the correlation between the hyperspectral characteristic parameters of honeydew pomelo leaves and relative chlorophyll content (SPAD) [48]. At the same time, this paper found that blindly increasing the amount of nitrogen application is not conducive to plant growth, while a reasonable amount of nitrogen application is conducive to improving the yield. Tao Sun conducted field trials in Jinhua City, Zhejiang Province, for three consecutive growing seasons. Two japonica-candid hybrids and two japonica rice varieties were treated with five nitrogen application rates (0, 150, 225, 300, and 375 kg/666 m$^2$). The amount of nitrogen increased, and the amount of nitrogen uptake in each growth period increased significantly with the increase in the amount of nitrogen applied [49].However, according to Rajesh, he conducted field tests during 2011–2012, and excessive use of fertilizer will not benefit nitrogen absorption. There is no linear increase with the amount of nitrogen used, and the higher nitrogen content shows a significantly reduced the NUE value [50]. Therefore, one of the goals of this article is to find an optimal nitrogen concentration for rice growth and to achieve efficient rice planting.

As important physiological indicators, whole leaf area, SPAD value, and the dry matter weight at points are of great significance for the rapid monitoring of crop breeding. As the most basic physiological index, the dry weight of crops directly indicates the accumulation of organic matter in crops, and the growth status of crops can be judged by combining the growth days of crops. Chlorophyll, as the main factory of plant photosynthesis, is an important indicator of concern to plant breeding workers. The whole leaf area of a plant represents the z of the light energy received by the plant. However, due to the lack of effective and rapid monitoring means, it is difficult to achieve large-scale rapid monitoring.

With the development of space technology, remote sensing plays an important role in modern agriculture, enabling people to monitor the growing distance of crops in the short and long term. Liu et al. used ESA Sentry-2 multi-spectral images and US Geological Survey DEM to combine object-oriented image analysis and DEM-based flow network analysis to map and to quantify the aquatic habitats of lentils [51]. However, satellite remote sensing technology often has shortcomings, such as a lack of accuracy. It is difficult to obtain data at a centimeter resolution. Although it is possible to accurately monitor the physiological status of plants in the field in real time by establishing a continuous monitoring system through small field stations, it is difficult to implement it on a large scale due to the high cost of establishment and complex maintenance. UAV is flexible in taking off and landing, is less affected by the environment, and has the advantage of being able to efficiently obtain spectral information. Therefore, we used DJI Phantom IV UAV to carry a multi-spectral platform to obtain high-resolution multi-spectral data, which makes the test results more accurate.

The traditional linear regression model has the advantage of a simple model and easy operation, but it is more difficult to construct the vegetation index spectrum, and it is difficult to meet the higher-level accuracy statistical regression. With the upgrade of computer hardware, recent machine learning methods adopt a nonlinear regression method, and this is widely used in agricultural monitoring by virtue of its excellent fitting ability. Other researchers around the world have also achieved good results through machine learning models. Guo et al. used machine learning methods, such as the backpropagation neural network model BP, support vector machine, random forest, and extreme learning machine, to predict corn yield [52]. The mean absolute errors of the BP, SVM, RF, and ELM models are 1.739, 0.886, 0.925, and 1.356, respectively. Zhang et al. used low-cost UV-derived photographic point clouds and machine learning methods to map the canopy heights of dense tropical forests [53]. Kim et al. used a multi-spectral camera on a drone to

detect weed spots in a buckwheat field [54]. However, there has been a lack of research on the comparison of machine learning models on rice. We introduced machine learning models into experiments and compared them, which makes the research more practical.

Given that the variation of various plant physiological parameters could induce strong responses in specific vegetation indexes, the unbalanced nature of the spectral features selected from the different vegetation indices could be attributed to the sensitivity of different physiological parameters. Many previous studies have proven the correlation between the parameters of remote sensing images extracted by UAV and plant chlorophyll. A recent study conducted by Fang et al. also confirms that NDVI can be used to detect plant chlorophyll [55]. In order to explore the deep relationship between different vegetation indices and physiological parameters, this paper uses multiple machine learning models to establish the relationship model between vegetation indices and physiological parameters. The results show a strong correlation between the physiological index and the vegetation index. Dry matter is predicted via GNDVI through a regression model, and the $R^2$ between the predicted value and the real value is as high as 0.967. There is also a good correlation between whole leaf area and SPAD value, which proves that there is a direct relationship between the vegetation index and the physiological index. In the case of a single independent variable, compared with other vegetation indices, EVI, NDVI, and RVI have higher accuracies and they can better predict the plant physiological parameters.

Due to a single vegetation index containing less information, it is difficult to comprehensively reflect the growth of crops. When a single vegetation index is used to establish the model, although the correlation is verified, the accuracy of the model is low. Correspondingly, combining multiple vegetation indexes can effectively obtain more information about plants, greatly improving the accuracy of the model. As a result, using multiple vegetation indices to predict the physiological indices will have a better effect. A recent study conducted by Qi et al. also confirms that it is better to use multiple vegetation indices to predict physiological parameters than to use one vegetation index [14]. This paper uses machine learning models to simulate the relationship between the physiological indices of three plants and nine different vegetation indices, and as the evaluation index to monitor the degree of fit of the linear model of rice leaf SPAD value and MSE, RMSE, MAE, MAPE, and SMAPE are used as the evaluation criteria of model accuracy. The results show that the effect of using multiple vegetation indexes to predict chlorophyll concentration is significantly better than using a single vegetation index, and it is of significance to continue the research. However, the prediction results of the whole leaf area and dry matter accumulation are not ideal, so it is difficult to predict this accurately. When comparing four different machine learning models, the random forest model has the best effect. The best univariate regression model and multivariate regression model both use the random forest model. As an important model of machine learning, the random forest and SVM have many advantages. The model introduces randomness, with adequate anti-noise performance and a high accuracy rate, which can better complete data fitting.

This work provided a machine learning system for monitoring rice physiological parameters using aerial drone images. The main focus of our study is on monitoring rice physiology. This method is not only helpful for monitoring crop growth in the field, but also for several other applications worth mentioning, including quickly collecting plant traits and screening excellent varieties. We innovatively used the multi-vegetation index, combined with the machine learning algorithm. We compared the difference between the single vegetation index and the multi-vegetation index on the prediction results of the model. We found that it is difficult to comprehensively evaluate the overall physiology of rice with a single vegetation index. Comparing with the single vegetation index model, relationships include other vegetation indexes, and they coincide with the field data observed on freeways, which greatly improves the prediction results. This model realizes the rapid detection of rice field growth, and provides a new idea for UAVs to monitor crop growth in the field.

## 5. Conclusions

In this work, a new, advanced solution for the prediction of rice physiological parameters by acquiring multi-spectral images through multi-spectral equipment carried by UAVs is presented. This research made full use of the abundant signal, strong observation ability, and the real-time characteristics of multi-spectral images, and analyzed the spectral characteristics of different rice varieties, nitrogen application levels, and growth periods. Finally, a multi-spectral image–rice physiological index model was constructed. Moreover, the relationships between nitrogen application and rice growth were analyzed and discussed. The main conclusions can be drawn as follows:

First, the multi-vegetation index composed of nine vegetation indexes was used as the input parameter. Compared with single vegetation, the multi-vegetation index has a better effect in MSE and MAPE. Then, we compared the different machine learning models, and we found that SVM and RF were used to establish the model, and that the overall prediction effect was good, which significantly improved its ability to monitor rice physiological parameters, and effectively reduced its dependence on test conditions such as variety and soil fertility.

Second, the analysis shows that increasing the amount of nitrogen application can promote rice growth, and increase the chlorophyll concentration, dry matter, and whole leaf area of rice in the early stage of rice growth, but excessive nitrogen application will not even inhibit rice growth.

The machine learning model showed a certain degree of retrieval accuracy for the study area. Nevertheless, the spectral inversion model is influenced by many factors, including light sources, temperature, weather conditions, etc. Additionally, compared with other studies, high-resolution multi-spectral images were adopted in this study. Therefore, the differences in the resolution of images, types of aircraft, and multi-spectral camera bands may affect the retrieval results. In the actual application, the model should be selected based on the development of the model, using the accuracy requirements and the results of current surveys. At the same time, because the experimental location is limited to Guangdong Province, China, the model is not practical for other regions. Subsequent experiments can be conducted in different regions to improve the universality of the model.

**Author Contributions:** Conceptualization, S.L., T.C. and L.Z.; Data curation, W.Y., Y.G. and S.Y.; Formal analysis, S.L. and Y.L. (Yongda Lin); Funding acquisition, L.Z.; Investigation, S.L., B.Z., H.Z., J.T. and X.L.; Methodology, S.L., W.Y. and L.Z.; Project administration, B.Z., Y.L. (Yubin Lan) and L.Z.; Resources, B.Z. and H.Z.; Software, S.L., W.Y., T.C. and L.Z.; Validation, Y.L. (Yongda Lin) and X.L.; Visualization, T.C. and J.T.; Writing—original draft, S.L., B.Z. and W.Y.; Writing—review and editing, T.C. and L.Z. All authors have read and agreed to the published version of the manuscript.

**Funding:** This research was funded by the Key Science and Technology Planning Project of Guangdong Province (2019B020214003), the National Key R&D Program of China (2020YFD1000905), the Guangdong Technical System of Peanut and Soybean Industry (2022KJ136-05), and the National Natural Science Foundation of China (42005142).

**Data Availability Statement:** The data provided in this study can be made available upon request from the corresponding author. The data have not been made public because they are still being used for further research.

**Conflicts of Interest:** The authors declare no conflict of interest.

## Appendix A

**Table A1.** Analysis of Variance of Physiological Parameters of Rice.

| Varieties | Reproductive Period | Nitrogen Application Level | SPAD | Dry Matter Weight (kg/666 m²) | Whole Leaf Area (cm²) |
|---|---|---|---|---|---|
| Meixiangzhan 2 | Tillering Stage | N1 | 33.79 ± 0.81 Aa | 239.50 ± 20.10 Abcd | 1105.89 ± 189.30 Aa |
| | | N2 | 35.97 ± 0.22 Ba | 237.20 ± 9.41 Aa | 1368.00 ± 49.80 Aab |
| | | N3 | 35.98 ± 1.16 Ba | 248.80 ± 17.28 Aab | 1514.91 ± 150.92 Aab |
| | | N4 | 36.50 ± 0.39 Bab | 280.40 ± 31.56 Aabc | 1520.47 ± 169.36 Aa |
| | Full Heading Stage | N1 | 30.40 ± 0.39 Aa | 628.45 ± 40.35 Aa | 1581.88 ± 79.55 Abc |
| | | N2 | 31.35 ± 0.034 Aa | 679.14 ± 28.84 Aab | 1950.26 ± 241.32 Ba |
| | | N3 | 32.77 ± 0.80 Ba | 657.05 ± 41.00 Aa | 2075.52 ± 61.18 BCab |
| | | N4 | 31.33 ± 0.66 Aa | 663.74 ± 49.94 Aa | 2347.00 ± 60.42 Cab |
| Ivory Xiangzhan | Tillering Stage | N1 | 36.12 ± 1.26 Ac | 239.30 ± 5.32 Abcd | 1026.09 ± 43.45 Aa |
| | | N2 | 36.63 ± 1.44 Aab | 259.30 ± 17.77 ABab | 1348.17 ± 132.63 Bab |
| | | N3 | 37.27 ± 0.75 Aa | 287.50 ± 17.49 Bbcd | 1582.71 ± 131.31 Bab |
| | | N4 | 36.37 ± 1.26 Aab | 262.50 ± 12.26 Abab | 1499.07 ± 93.55 Ba |
| | Full Heading Stage | N1 | 29.53 ± 0.43 Aa | 618.79 ± 69.91 Aa | 1418.39 ± 109.60 Aabc |
| | | N2 | 30.33 ± 0.62 Aa | 633.78 ± 32.67 Aab | 1942.61 ± 117.91 Ba |
| | | N3 | 33.47 ± 0.88 Ba | 656.48 ± 44.94 Aa | 1814.85 ± 362.83 ABa |
| | | N4 | 30.93 ± 0.28 Aa | 667.60 ± 39.24 Aa | 1969.68 ± 120.58 Ba |
| 19 Xiang | Tillering Stage | N1 | 35.55 ± 0.75 Aabc | 220.60 ± 27.75 Abc | 981.31 ± 156.31 Aa |
| | | N2 | 36.77 ± 0.79 ABab | 276.30 ± 17.83 Bab | 1261.80 ± 93.41 ABab |
| | | N3 | 37.35 ± 0.73 Ba | 233.30 ± 17.10 ABa | 1290.97 ± 128.16 Ba |
| | | N4 | 37.02 ± 0.11 ABab | 251.00 ± 13.34 ABa | 1304.23 ± 118.67 Ba |
| | Full Heading Stage | N1 | 35.20 ± 0.63 Ae | 636.16 ± 65.84 Aa | 1345.31 ± 228.15 Aab |
| | | N2 | 37.06 ± 0.42 Bd | 736.78 ± 28.50 Abc | 2049.05 ± 67.37 Ba |
| | | N3 | 37.70 ± 0.56 Bc | 681.05 ± 82.11 Aa | 2176.54 ± 232.57 Bab |
| | | N4 | 38.08 ± 0.19 Bd | 819.92 ± 16.708 Bb | 2355.12 ± 286.83 Bab |
| Ruanhuayou Jinsi | Tillering Stage | N1 | 34.03 ± 0.77 Aab | 250.30 ± 17.94 Acd | 1079.60 ± 89.80 Aa |
| | | N2 | 37.17 ± 0.62 Bab | 284.60 ± 30.78 Abb | 1327.44 ± 189.90 Aab |
| | | N3 | 37.57 ± 0.38 Ba | 328.90 ± 38.85 Bd | 1707.20 ± 225.63 Bb |
| | | N4 | 36.89 ± 1.53 Bab | 308.10 ± 9.55 ABc | 1388.70 ± 110.63 ABa |
| | Full Heading Stage | N1 | 30.93 ± 0.28 Aa | 556.68 ± 17.49 Aa | 1164.75 ± 55.75 Aa |
| | | N2 | 32.81 ± 0.88 Bb | 747.33 ± 58.59 Bbc | 2076.61 ± 55.87 Ba |
| | | N3 | 35.31 ± 0.52 Cb | 805.91 ± 36.51 Bb | 2491.95 ± 323.08 Bb |
| | | N4 | 35.62 ± 0.80 Cb | 727.56 ± 30.93 Bab | 2151.58 ± 238.94 Bab |
| Qingxiangyou 033 | Tillering Stage | N1 | 35.26 ± 1.14 Aabc | 261.80 ± 13.52 Ad | 1060.26 ± 70.00 Aa |
| | | N2 | 38.06 ± 0.85 Bab | 283.30 ± 15.31 ABb | 1446.56 ± 84.78 Bb |
| | | N3 | 37.26 ± 0.95 Aba | 324.70 ± 14.64 Cd | 1465.37 ± 51.39 Bab |
| | | N4 | 37.01 ± 1.32 ABab | 302.80 ± 19.94 BCbc | 1421.53 ± 39.42 Ba |
| | Full Heading Stage | N1 | 31.85 ± 0.85 Abc | 594.13 ± 8.52 Aa | 1762.00 ± 365.02 Ac |
| | | N2 | 34.07 ± 0.57 Bbc | 674.58 ± 82.49 (A,ab) | 2128.03 ± 261.29 Aa |
| | | N3 | 33.95 ± 0.84 Ba | 649.02 ± 22.03 (A,a) | 2040.83 ± 272.11 Aab |
| | | N4 | 36.60 ± 0.19 Cbc | 671.64 ± 21.51 (A,a) | 2037.58 ± 89.11 Aa |
| Nanjingxiangzhan | Tillering Stage | N1 | 33.90 ± 0.80 Aa | 209.90 ± 17.6155 (A,b) | 1015.77 ± 89.23 Aa |
| | | N2 | 36.59 ± 1.46 Bab | 241.80 ± 15.35 (AB,ab) | 1203.73 ± 118.34 Aab |
| | | N3 | 36.65 ± 1.32 Ba | 271.50 ± 13.81 (B,abc) | 1438.87 ± 67.69 Bab |
| | | N4 | 35.81 ± 0.32 ABab | 277.50 ± 20.85 (B,abc) | 1473.45 ± 99.02 Ba |
| | Full Heading Stage | N1 | 32.82 ± 0.65 Acd | 545.54 ± 17.46 (A,a) | 1411.14 ± 112.87 Aabc |
| | | N2 | 33.91 ± 0.86 Abc | 704.81 ± 21.53 (B,ab) | 2065.37 ± 217.53 Ba |
| | | N3 | 36.22 ± 0.50 Bb | 658.77 ± 29.27 (AB,a) | 1908.01 ± 128.49 Ba |
| | | N4 | 37.21 ± 0.34 Bcd | 644.48 ± 92.08 (AB,a) | 1992.98 ± 107.71 Ba |
| Erguangxiangzhan 3 | Tillering Stage | N1 | 36.73 ± 1.85 Ac | 212.40 ± 15.00 (A,bc) | 991.43 ± 58.72 Aa |
| | | N2 | 38.33 ± 0.83 Ab | 269.00 ± 13.79 (B,ab) | 1348.92 ± 61.167 Bab |
| | | N3 | 37.31 ± 0.33 Aa | 307.30 ± 9.54 (C,cd) | 1526.36 ± 48.74 Cab |
| | | N4 | 39.25 ± 0.80 Ac | 251.00 ± 18.17 Ba | 1355.34 ± 49.75 Ba |
| | Full Heading Stage | N1 | 36.25 ± 0.18 Ae | 606.01 ± 45.18 Aa | 1280.40 ± 54.35 Aab |
| | | N2 | 38.15 ± 0.74 Bd | 828.62 ± 14.61 ABc | 2206.06 ± 173.17 Ba |
| | | N3 | 39.22 ± 0.55 Bd | 711.63 ± 42.87 ABab | 2068.16 ± 24.20 Bab |
| | | N4 | 41.85 ± 0.88 Ce | 668.51 ± 89.13 Aa | 2001.28 ± 159.28 Ba |

**Table A1.** *Cont.*

| Varieties | Reproductive Period | Nitrogen Application Level | SPAD | Dry Matter Weight (kg/666 m²) | Whole Leaf Area (cm²) |
|---|---|---|---|---|---|
| Lixiangzhan 4 | Tillering Stage | N1 | 35.82 ± 0.74 Abc | 233.20 ± 23.89 Aa | 1001.01 ± 72.31 Aa |
| | | N2 | 36.30 ± 0.40 Aab | 235.00 ± 17.79 Aa | 1187.70 ± 28.84 Ba |
| | | N3 | 36.90 ± 0.10 Aa | 317.20 ± 13.25 Bd | 1530.21 ± 63.41 Cab |
| | | N4 | 38.50 ± 0.65 Bbc | 278.90 ± 15.08 Babc | 1506.64 ± 77.22 Ca |
| | Full Heading Stage | N1 | 33.62 ± 1.12 Ad | 625.53 ± 13.03 Aa | 1625.80 ± 139.07 Abc |
| | | N2 | 34.37 ± 0.56 Ac | 675.57 ± 25.10 Aab | 2125.90 ± 210.50 Aa |
| | | N3 | 35.40 ± 0.21 Ab | 602.42 ± 69.71 Aa | 2176.20 ± 300.37 Bab |
| | | N4 | 35.65 ± 1.07 Ab | 640.96 ± 49.20 Aa | 2513.50 ± 118.29 Bb |

The capital letters in the table represent the variance analysis of different nitrogen application levels of peanuts at the same variety and growth period. The lowercase letters in the table represent the variance analysis for different varieties of peanuts with the same nitrogen application level and growth period.

## Appendix B

**Table A2.** Comparison of Prediction Accuracy of Complete Single-Factor Regression Models.

| Vegetation Index | Type | Model Type | MSE | RMSE | MAE | MAPE | SMAPE |
|---|---|---|---|---|---|---|---|
| DVI | SPAD | Linear regression | 4.44 | 2.11 | 1.54 | 4.41 | 4.38 |
| DVI | SPAD | Random tree | 3.29 | 1.81 | 1.35 | 6.60 | 6.54 |
| DVI | SPAD | Random forest | 1.62 | 1.27 | 0.83 | 7.00 | 6.99 |
| DVI | SPAD | SVM | 4.40 | 2.10 | 1.48 | 6.07 | 5.98 |
| EVI | SPAD | Linear regression | 4.20 | 2.05 | 1.50 | 4.30 | 4.29 |
| EVI | SPAD | Random tree | 2.79 | 1.67 | 1.20 | 6.67 | 6.63 |
| EVI | SPAD | Random forest | 0.99 | 1.00 | 0.74 | 7.11 | 7.05 |
| EVI | SPAD | SVM | 3.92 | 1.98 | 1.42 | 6.17 | 6.10 |
| GNDVI | SPAD | Linear regression | 4.79 | 2.19 | 1.61 | 4.61 | 4.59 |
| GNDVI | SPAD | Random tree | 3.61 | 1.90 | 1.25 | 6.49 | 6.42 |
| GNDVI | SPAD | Random forest | 1.04 | 1.02 | 0.72 | 6.91 | 6.86 |
| GNDVI | SPAD | SVM | 4.73 | 2.18 | 1.59 | 6.09 | 6.01 |
| LCI | SPAD | Linear regression | 3.91 | 1.98 | 1.45 | 4.14 | 4.13 |
| LCI | SPAD | Random tree | 2.21 | 1.49 | 1.03 | 6.82 | 6.78 |
| LCI | SPAD | Random forest | 0.68 | 0.82 | 0.57 | 7.31 | 7.25 |
| LCI | SPAD | SVM | 3.25 | 1.80 | 1.28 | 6.36 | 6.31 |
| NDVI | SPAD | Linear regression | 4.10 | 2.02 | 1.48 | 4.22 | 4.21 |
| NDVI | SPAD | Random tree | 2.97 | 1.72 | 1.28 | 6.69 | 6.64 |
| NDVI | SPAD | Random forest | 1.00 | 1.00 | 0.72 | 6.83 | 6.76 |
| NDVI | SPAD | SVM | 3.82 | 1.96 | 1.39 | 6.19 | 6.12 |
| NLI | SPAD | Linear regression | 4.49 | 2.12 | 1.52 | 4.36 | 4.33 |
| NLI | SPAD | Random tree | 3.66 | 1.91 | 1.35 | 6.56 | 6.48 |
| NLI | SPAD | Random forest | 1.31 | 1.14 | 0.75 | 6.82 | 6.76 |
| NLI | SPAD | SVM | 4.52 | 2.13 | 1.52 | 5.98 | 5.88 |
| OSAVI | SPAD | Linear regression | 4.10 | 2.02 | 1.48 | 4.22 | 4.21 |
| OSAVI | SPAD | Random tree | 2.97 | 1.72 | 1.28 | 6.69 | 6.64 |
| OSAVI | SPAD | Random forest | 1.13 | 1.06 | 0.75 | 6.88 | 6.79 |
| OSAVI | SPAD | SVM | 3.82 | 1.96 | 1.39 | 6.19 | 6.12 |
| RVI | SPAD | Linear regression | 4.55 | 2.13 | 1.57 | 4.50 | 4.48 |
| RVI | SPAD | Random tree | 2.97 | 1.72 | 1.28 | 6.69 | 6.64 |
| RVI | SPAD | Random forest | 1.15 | 1.07 | 0.70 | 6.84 | 6.80 |
| RVI | SPAD | SVM | 4.55 | 2.13 | 1.55 | 6.14 | 6.07 |
| SAVI | SPAD | Linear regression | 4.10 | 2.02 | 1.48 | 4.22 | 4.21 |
| SAVI | SPAD | Random tree | 2.97 | 1.72 | 1.28 | 6.69 | 6.64 |
| SAVI | SPAD | Random forest | 1.16 | 1.08 | 0.74 | 7.11 | 7.03 |
| SAVI | SPAD | SVM | 3.83 | 1.96 | 1.39 | 6.19 | 6.12 |

**Table A2.** *Cont.*

| Vegetation Index | Type | Model Type | MSE | RMSE | MAE | MAPE | SMAPE |
|---|---|---|---|---|---|---|---|
| SIPS2 | SPAD | Linear regression | 5.95 | 2.44 | 1.93 | 5.58 | 5.49 |
| SIPS2 | SPAD | Random tree | 3.64 | 1.91 | 1.29 | 6.36 | 6.30 |
| SIP2 | SPAD | Random forest | 1.49 | 1.22 | 0.79 | 6.55 | 6.46 |
| SIPI2 | SPAD | SVM | 4.79 | 2.19 | 1.60 | 5.80 | 5.67 |
| DVI | Dry matter accumulation | Linear regression | 25,272.66 | 158.97 | 130.24 | 35.04 | 28.83 |
| DVI | Dry matter accumulation | Random tree | 15,235.54 | 123.43 | 81.44 | 56.67 | 46.47 |
| DVI | Dry matter accumulation | Random forest | 6118.24 | 78.22 | 48.79 | 57.85 | 47.21 |
| DVI | Dry matter accumulation | SVM | 40,626.66 | 201.56 | 192.42 | 51.03 | 45.44 |
| EVI | Dry matter accumulation | Linear regression | 12,799.01 | 113.13 | 82.80 | 23.16 | 19.28 |
| EVI | Dry matter accumulation | Random tree | 1718.21 | 41.45 | 27.01 | 59.13 | 48.31 |
| EVI | Dry matter accumulation | Random forest | 669.17 | 25.87 | 16.29 | 59.19 | 48.55 |
| EVI | Dry matter accumulation | SVM | 36,594.71 | 191.30 | 181.56 | 51.02 | 45.42 |
| GNDVI | Dry matter accumulation | Linear regression | 3028.99 | 55.04 | 40.62 | 10.03 | 9.80 |
| GNDVI | Dry matter accumulation | Random tree | 1164.05 | 34.12 | 24.37 | 59.19 | 48.30 |
| GNDVI | Dry matter accumulation | Random forest | 661.81 | 25.73 | 17.81 | 59.43 | 48.54 |
| GNDVI | Dry matter accumulation | SVM | 32,760.81 | 181.00 | 170.95 | 50.66 | 45.40 |
| LCI | Dry matter accumulation | Linear regression | 16,640.53 | 129.00 | 97.03 | 27.92 | 23.57 |
| LCI | Dry matter accumulation | Random tree | 4586.09 | 67.72 | 44.55 | 58.47 | 47.77 |
| LCI | Dry matter accumulation | Random forest | 3798.58 | 61.63 | 34.35 | 58.46 | 47.58 |
| LCI | Dry matter accumulation | SVM | 37,997.51 | 194.93 | 185.09 | 50.98 | 45.43 |
| NDVI | Dry matter accumulation | Linear regression | 11,372.35 | 106.64 | 78.45 | 20.16 | 17.40 |
| NDVI | Dry matter accumulation | Random tree | 1753.56 | 41.88 | 27.49 | 59.16 | 48.34 |
| NDVI | Dry matter accumulation | Random forest | 760.43 | 27.58 | 16.20 | 58.57 | 48.25 |
| NDVI | Dry matter accumulation | SVM | 35,972.30 | 189.66 | 180.11 | 50.98 | 45.42 |
| NLI | Dry matter accumulation | Linear regression | 22,520.33 | 150.07 | 121.80 | 30.74 | 26.16 |
| NLI | Dry matter accumulation | Random tree | 13,801.87 | 117.48 | 69.99 | 56.61 | 46.31 |
| NLI | Dry matter accumulation | Random forest | 6512.88 | 80.70 | 49.70 | 57.12 | 47.02 |
| NLI | Dry matter accumulation | SVM | 39,097.22 | 197.73 | 188.70 | 51.23 | 45.43 |
| OSAVI | Dry matter accumulation | Linear regression | 11,388.91 | 106.72 | 78.52 | 20.18 | 17.41 |
| OSAVI | Dry matter accumulation | Random tree | 1753.56 | 41.88 | 27.49 | 59.16 | 48.34 |
| OSAVI | Dry matter accumulation | Random forest | 558.03 | 23.62 | 15.42 | 59.36 | 48.41 |
| OSAVI | Dry matter accumulation | SVM | 35,978.27 | 189.68 | 180.13 | 50.98 | 45.42 |
| RVI | Dry matter accumulation | Linear regression | 10,965.51 | 104.72 | 72.97 | 22.22 | 19.04 |
| RVI | Dry matter accumulation | Random tree | 1768.99 | 42.06 | 27.07 | 59.12 | 48.30 |
| RVI | Dry matter accumulation | Random forest | 615.61 | 24.81 | 15.74 | 58.77 | 48.37 |
| RVI | Dry matter accumulation | SVM | 35,824.93 | 189.27 | 179.53 | 51.04 | 45.42 |
| SAVI | Dry matter accumulation | Linear regression | 11,423.99 | 106.88 | 78.66 | 20.22 | 17.45 |
| SAVI | Dry matter accumulation | Random tree | 1753.56 | 41.88 | 27.49 | 59.16 | 48.34 |
| SAVI | Dry matter accumulation | Random forest | 557.48 | 23.61 | 15.21 | 59.21 | 48.58 |
| SAVI | Dry matter accumulation | SVM | 35,990.92 | 189.71 | 180.16 | 50.98 | 45.42 |
| SIPS2 | Dry matter accumulation | Linear regression | 39,057.18 | 197.63 | 185.62 | 49.58 | 42.86 |
| SIPS2 | Dry matter accumulation | Random tree | 1409.91 | 37.55 | 28.18 | 59.07 | 48.20 |
| SIP2 | Dry matter accumulation | Random forest | 567.53 | 23.82 | 16.29 | 60.00 | 48.75 |
| SIPI2 | Dry matter accumulation | SVM | 39,490.60 | 198.72 | 189.85 | 50.79 | 45.44 |
| DVI | Whole leaf area | Linear regression | 165,290.47 | 406.56 | 338.79 | 22.50 | 20.84 |
| DVI | Whole leaf area | Random tree | 123,097.42 | 350.85 | 248.08 | 25.74 | 24.21 |
| DVI | Whole leaf area | Random forest | 29,272.65 | 171.09 | 120.38 | 27.75 | 25.88 |
| DVI | Whole leaf area | SVM | 182,313.12 | 426.98 | 347.75 | 21.47 | 21.65 |
| EVI | Whole leaf area | Linear regression | 130,033.38 | 360.60 | 282.98 | 19.21 | 17.43 |
| EVI | Whole leaf area | Random tree | 27,643.13 | 166.26 | 121.24 | 29.65 | 27.95 |
| EVI | Whole leaf area | Random forest | 15,666.32 | 125.17 | 87.33 | 29.67 | 27.82 |
| EVI | Whole leaf area | SVM | 176,291.82 | 419.87 | 342.79 | 21.63 | 21.70 |
| GNDVI | Whole leaf area | Linear regression | 79,538.29 | 282.03 | 220.06 | 14.91 | 13.92 |
| GNDVI | Whole leaf area | Random tree | 17,395.32 | 131.89 | 94.15 | 29.96 | 28.22 |
| GNDVI | Whole leaf area | Random forest | 9462.54 | 97.28 | 64.36 | 30.24 | 28.36 |
| GNDVI | Whole leaf area | SVM | 169,213.80 | 411.36 | 336.42 | 21.73 | 21.73 |

**Table A2.** *Cont.*

| Vegetation Index | Type | Model Type | MSE | RMSE | MAE | MAPE | SMAPE |
|---|---|---|---|---|---|---|---|
| LCI | Whole leaf area | Linear regression | 134,879.89 | 367.26 | 284.19 | 19.28 | 17.45 |
| LCI | Whole leaf area | Random tree | 71,123.41 | 266.69 | 187.69 | 27.74 | 26.05 |
| LCI | Whole leaf area | Random forest | 28,645.13 | 169.25 | 101.89 | 28.00 | 26.34 |
| LCI | Whole leaf area | SVM | 178,639.35 | 422.66 | 345.00 | 21.59 | 21.69 |
| NDVI | Whole leaf area | Linear regression | 128,002.80 | 357.77 | 280.84 | 18.89 | 17.26 |
| NDVI | Whole leaf area | Random tree | 26,831.71 | 163.80 | 119.40 | 29.71 | 28.01 |
| NDVI | Whole leaf area | Random forest | 12,495.40 | 111.78 | 68.31 | 30.28 | 28.53 |
| NDVI | Whole leaf area | SVM | 174,517.98 | 417.75 | 341.06 | 21.65 | 21.71 |
| NLI | Whole leaf area | Linear regression | 161,138.78 | 401.42 | 329.63 | 21.86 | 20.22 |
| NLI | Whole leaf area | Random tree | 103,305.31 | 321.41 | 210.89 | 26.55 | 24.98 |
| NLI | Whole leaf area | Random forest | 44,411.55 | 210.74 | 126.89 | 27.68 | 25.77 |
| NLI | Whole leaf area | SVM | 178,843.23 | 422.90 | 344.64 | 21.57 | 21.68 |
| OSAVI | Whole leaf area | Linear regression | 128,074.73 | 357.88 | 280.96 | 18.90 | 17.27 |
| OSAVI | Whole leaf area | Random tree | 26,831.71 | 163.80 | 119.40 | 29.71 | 28.01 |
| OSAVI | Whole leaf area | Random forest | 11,681.40 | 108.08 | 70.51 | 29.03 | 27.51 |
| OSAVI | Whole leaf area | SVM | 174,529.22 | 417.77 | 341.07 | 21.65 | 21.71 |
| RVI | Whole leaf area | Linear regression | 113,690.16 | 337.18 | 268.98 | 18.63 | 16.85 |
| RVI | Whole leaf area | Random tree | 26,831.71 | 163.80 | 119.40 | 29.71 | 28.01 |
| RVI | Whole leaf area | Random forest | 11,729.30 | 108.30 | 71.34 | 29.49 | 27.91 |
| RVI | Whole leaf area | SVM | 174,965.33 | 418.29 | 341.81 | 21.65 | 21.71 |
| SAVI | Whole leaf area | Linear regression | 128,226.56 | 358.09 | 281.22 | 18.92 | 17.29 |
| SAVI | Whole leaf area | Random tree | 26,831.71 | 163.80 | 119.40 | 29.71 | 28.01 |
| SAVI | Whole leaf area | Random forest | 7346.30 | 85.71 | 60.75 | 29.92 | 28.27 |
| SAVI | Whole leaf area | SVM | 174,553.00 | 417.80 | 341.08 | 21.65 | 21.71 |
| SIPS2 | Whole leaf area | Linear regression | 153,025.81 | 391.19 | 329.70 | 21.69 | 20.45 |
| SIPS2 | Whole leaf area | Random tree | 34,384.50 | 185.43 | 141.18 | 29.20 | 27.46 |
| SIP2 | Whole leaf area | Random forest | 11,312.49 | 106.36 | 79.31 | 29.11 | 27.41 |
| SIPI2 | Whole leaf area | SVM | 179,197.60 | 423.32 | 344.84 | 21.47 | 21.65 |

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
