# Peer review of "Quantification of Physiological Parameters of Rice Varieties Based on Multi-Spectral Remote Sensing and Machine Learning Models"

_remotesensing, doi:10.3390/rs15020453_

Round 1
Reviewer 1 Report
Monitoring method for physiological parameters of rice varieties based on multi-spectral remote sensing
Dear Authors
The basic science of this paper is conducted in a good way and is of an appropriate standard. The author and his team write this paper according to journal scope and modern. The author used UAV image data is relatively complex and difficult to analyze, which is the main problem limiting large-scale use at present. For plant physiological indicators from multispectral observations, this study describes a novel method, which is based on machine learning. Using UAV for deriving the absorption coefficient of plant canopies, we quantify the effect of reflectance spectra on the relationship between the plant physiological indicators such as the chlorophyll content, whole leaf area, and dry matter accumulation with nine spectral indices. The author also used the SVM model to predict the SPAD values of plants.
I have some minor and major comments.
Line 23: the author already explained the abbreviation in line 20.
Line 28: What is SPAD?
The author should explain the abbreviation the first time.
Some sentences are very long. I can’t understand clearly.
Line 65-99: This paragraph is very complex and I can’t find the research questions in this paragraph. The author just discusses the UAV in different crop stages.
The author should add the flow chart in the material and method section. It's not appropriate according to journal criteria.
Objectives are not clear at the end of the introduction section.
Figure 1: add after figure 2
Add more details of the experimental site.
Line 150 and 151: check the typo mistake.
Results in a very good form. I agree about the results.
Figure 4 export in high resolution according to journal criteria
The conclusion section is very weak. The author should modify the conclusion section.
There are many typos mistakes in this whole manuscript.
The author should check the whole manuscript.
Best Regards
Author Response
Journal Name: Remote Sensing
Manuscript ID: remotesensing-2086518
Manuscript Title: Quantification of physiological parameters of rice varieties based on multi-spectral remote sensing and Machine Learning models
Dear Editor,
Thank you for your letter and careful consideration of our manuscript remotesensing-2086518 entitled “Quantification of physiological parameters of rice varieties based on multi-spectral remote sensing and Machine Learning models”. We appreciate the opportunity to revise and resubmit our manuscript. The comments were extremely valuable and have helped us to improve our manuscript greatly. We have modified the manuscript accordingly and hoped the revised manuscript would meet your requests. And then detailed corrections are listed below point by point.
Comments :
1: Line 23: the author already explained the abbreviation in line 20.
Responses:
ü Thank you for your comments. I have deleted the Duplicate content.
2: Line 28: What is SPAD? The author should explain the abbreviation the first time.
Responses:
ü Thank you for your comments. I have added the full name of SPAD.
Point 3: Some sentences are very long. I can’t understand clearly.
Response:
ü Thank you for your comments. After carefully check the article, the long sentences were spilt clear short sentences. And the paper is modified through the polishing mechanism recommended by MDPI.
Point 4: Line 65-99: This paragraph is very complex and I can’t find the research questions in this paragraph. The author just discusses the UAV in different crop stages.
Response:
ü Thank you for your comments. After carefully check the article, the section about UAV is deleted. Instead, the paper adds the research status of agricultural parameters inversion through machine learning, which is used to prove the rationality of machine learning for remote sensing.
Point 5: The author should add the flow chart in the material and method section. It's not appropriate according to journal criteria.
Response:
ü Thank you for your comments, I have added the flow chart in the material and method section.
Point 6: Objectives are not clear at the end of the introduction section.
Response:
ü Thank you for your comments, after carefully check the article, we re-edited the research of objectives. We cleared the research objective of this paper as further extend multispectral prediction model of rice physiological parameters through different growth stages, varieties, and nitrogen application. We also discussed the effect of nitrogen application rate on different rice varieties.
Point 7: Figure 1: add after figure 2
Response:
ü Thank you for your comments, the figure has been revised in the article.
Point 8: Add more details of the experimental site.
Response:
ü Thank you for your comments. We modified the materials and methods part of the article and added the detailed description of the experimental site.
Point 9: Line 150 and 151: check the typo mistake.
Response:
ü Thank you for your comments. The format problem has been modified
Point 10: Figure 4 export in high resolution according to journal criteria
Response:
ü Thank you for your comments. The picture has been modified.
Point 11: The conclusion section is very weak. The author should modify the conclusion section.
Response:
ü Thank you for your comments. After carefully check the articles, we have edited the conclusion of the article and sorted out the main purposes of this work. First, this work established a spectral image transformation model for rice physiological parameters, and then discussed the effects of different nitrogen fertilizer applications on rice growth and development. And the conclusion part adds the shortcomings of this work.
Point 12: There are many typos mistakes in this whole manuscript. The author should check the whole manuscript.
Response:
ü Thank you for your comments. The author checked the syntax of the full text in detail, made a lot of modifications to the unreasonable places, and polished the article through the polishing mechanism recommended by MDPI.
This revised manuscript has not been published or presented elsewhere in part or in entirety, and is not under consideration by another journal. All the authors have approved the manuscript and agree with submission to your esteemed journal. There are no conflicts of interest to declare. Thank you for your consideration. I look forward to hearing from you.
Sincerely,
Lei Zhang
College of Agriculture, South China Agricultural University,
Guangzhou, 510642, PR China
Phone: (86)-20-85280203
Fax: (86)-20-85280203
E-mail: zhanglei@scau.edu.cn

Reviewer 2 Report
Dear Authors,
The subject of the study is interesting and topical, with scientific and practical importance.
The introduction is presented correctly, in accordance with the subject. Numerous scientific articles, in concordance to the topic of the study, were consulted.
Methodology of the study was clearly presented, and appropriate to the proposed objectives.
The obtained results are important and have been analyzed and interpreted correctly, in accordance with the current methodology.
Some suggestions were made in the article.
The following aspects are brought to the attention of the authors.
1.
Space between the bibliographic source and the word preceding it
Eg
Page 2, row 49
“them [5].” instead of “them[5].”
Several suggestions were made in the article
2.
The sentence and figure 1 referred to are appropriate in chapter 2. Materials and methods
It is recommended to revise chapter 2. Materials and Methods and insert figure 1 in that chapter.
It is possible to require the renumbering of the figures in chapter 2, after the recommended correction.
3.
“m2”
“R2”
4.
It is recommended to use the text settings according to Instructions for Authors, and Microsoft Word template, Remote Sensing journal.
Styles: MDPI 3.1 text
Eg
Page 6, the first paragraph
5.
The table is outside the page settings
Eg.
Table2, page 6
The review is recommended, according to Instructions for Authors and Microsoft Word template, Remote Sensing journal.
6.
The number of the bibliographic source next to the cited author/authors
Eg
Page 2, row 83
It is recommended to check in the article, especially in the Discussions chapter
7.
Author Contributions
It is recommended to revise to be in accordance with Instructions for Authors, and Microsoft Word template, Remote Sensing journal.
“Conceptualization, X.X. and Y.Y.; methodology, X.X.; ...”
abbreviated form for the authors' names
8.
References
The entire References chapter needs to be revised to be consistent with the instructions for Authors, and Microsoft Word template, Remote Sensing journal.
“Author 1, A.B.; Author 2, C.D. Title of the article. Abbreviated Journal Name Year, Volume, page range.”
Eg.
“Mourad, R.; Jaafar, H.; Anderson, M.; Gao, F. Assessment of leaf area index models using harmonized Landsat and Sentinel-2 surface reflectance data over a semi-arid irrigated landscape. Remote Sens. 2020, 12(19), 3121. https: //doi.org/10. 3390/rs12193121.“
Instead of
“Mourad, R., Jaafar, H., Anderson, M., & Gao, F. Assessment of leaf area index models using harmonized land-sat and sentinel-2 surface reflectance data over a semi-arid irrigated landscape. Remote Sensing. 2020,12(19). https: //doi.org/10. 3390/rs12193121.“

Author Response
Journal Name: Remote Sensing
Manuscript ID: remotesensing-2086518
Manuscript Title: Quantification of physiological parameters of rice varieties based on multi-spectral remote sensing and Machine Learning models
Dear Editor,
Thank you for your letter and careful consideration of our manuscript remotesensing-2086518 entitled “Quantification of physiological parameters of rice varieties based on multi-spectral remote sensing and Machine Learning models”. We appreciate the opportunity to revise and resubmit our manuscript. The comments were extremely valuable and have helped us to improve our manuscript greatly. We have modified the manuscript accordingly and hoped the revised manuscript would meet your requests. And then detailed corrections are listed below point by point.
Comments :
1: Space between the bibliographic source and the word preceding it
Eg:
Page 2, row 49:“them [5].” instead of “them[5].”
Several suggestions were made in the article.
Response:
ü Thank you for your comments. I have checked the full text according to the comments of the reviewer and revised it.
2: The sentence and figure 1 referred to are appropriate in chapter 2. Materials and methods
It is recommended to revise chapter 2. Materials and Methods and insert figure 1 in that chapter.
It is possible to require the renumbering of the figures in chapter 2, after the recommended correction.
Response:
ü Thank you for your comments. Figure 1 and related text have been moved to chapter 2.
3: “m2”instead of“R2”
Response:
ü Thank you for your comments. It has been modified according to the comments of the reviewer.
4: It is recommended to use the text settings according to Instructions for Authors, and Microsoft Word template, Remote Sensing journal.
Styles: MDPI 3.1 text
Eg: Page 6, the first paragraph
Response:
ü Thank you for your comments. I have modified the text format according to the MDPI format.
5: The table is outside the page settings
Eg.
Table2, page 6
The review is recommended, according to Instructions for Authors and Microsoft Word template, Remote Sensing journal.
Response:
ü Thank you for your comments. I have modified the text format according to the MDPI format.
6: The number of the bibliographic source next to the cited author/authors
Eg
Page 2, row 83
It is recommended to check in the article, especially in the Discussions chapter
Response:
ü Thank you for your comments. I have modified the text format according to the MDPI format.
7: Author Contributions
It is recommended to revise to be in accordance with Instructions for Authors, and Microsoft Word template, Remote Sensing journal.
“Conceptualization, X.X. and Y.Y.; methodology, X.X.; ...”
abbreviated form for the authors' names
Response:
ü Thank you for your comments. I have modified the text format according to the MDPI format.
8: References
The entire References chapter needs to be revised to be consistent with the instructions for Authors, and Microsoft Word template, Remote Sensing journal.
“Author 1, A.B.; Author 2, C.D. Title of the article. Abbreviated Journal Name Year, Volume, page range.”
Eg.
“Mourad, R.; Jaafar, H.; Anderson, M.; Gao, F. Assessment of leaf area index models using harmonized Landsat and Sentinel-2 surface reflectance data over a semi-arid irrigated landscape. Remote Sens. 2020, 12(19), 3121. https: //doi.org/10. 3390/rs12193121.“
Instead of
“Mourad, R., Jaafar, H., Anderson, M., & Gao, F. Assessment of leaf area index models using harmonized land-sat and sentinel-2 surface reflectance data over a semi-arid irrigated landscape. Remote Sensing. 2020,12(19). https: //doi.org/10. 3390/rs12193121.“
Response:
ü Thank you for your comments. I have modified the text format according to the MDPI format.
This revised manuscript has not been published or presented elsewhere in part or in entirety, and is not under consideration by another journal. All the authors have approved the manuscript and agree with submission to your esteemed journal. There are no conflicts of interest to declare. Thank you for your consideration. I look forward to hearing from you.
Sincerely,
Lei Zhang
College of Agriculture, South China Agricultural University,
Guangzhou, 510642, PR China
Phone: (86)-20-85280203
Fax: (86)-20-85280203
E-mail: zhanglei@scau.edu.cn

Reviewer 3 Report
The experimental work presented in the Manuscript, entitled „ Monitoring method for physiological parameters of rice varieties based on multi-spectral remote sensing ". The article reports that using SVM to predict chlorophyll content has better effect because of better adaptation and higher accuracy than other models. This study suggests that multispectral data acquired by UAV can quickly estimate field physiological indicators and which has great potential in the pre-visual detection of chlorophyll content information in the field. At the same time, it can also be extended to the detection and inversion of other key variables of crops; there are several shortcomings and modifications that should be included in order to enhance the manuscript for the readers.
1- Title: please added in the title (the machine learning models) the title should be (Quantification physiological parameters of rice varieties based on multi-spectral remote sensing and Machine Learning models). It better to reflect the all information about the manuscript in the title.
Abstract
2- Line 21 to line 23.The sentence (For plant physiological indicators from multispectral observations….. …. based on the machine learning) from line 21 to 23 should be rephrased.
3- Line 24. Please do not use pronouns in the formulation of the sentence. For example, we quantify.
4- Line 24 to line 25. This sentence is wrong (we quantify the effect of reflectance spectra on the relationship between the plant physiological indicators such as the chlorophyll content, hole leaf area and dry matter accumulation). Spectral reflections are affected by the change in physiological characteristics but not affected in physiological characteristics. Please remove this sentence or rephrasing it.5- Line 25. Do you mean hole or whole?
6- Line 26. Please write the full name of the spectral indices (EVI, GNDVI, LCI, NDVI, NLI, OSAVI, RVI, SAVI, SIPI2) or remove them.
7- The best results of spectral indices should be added in abstract.
8- The best results of SVM should be added in abstract.
Introduction
9- Please add the references for the first sentence in introduction?
10- Line 57. Change [8,9] to [8,9].
11- Line 60 to line 63. References should be added from line 60 to line 63.
12- Line 68 to line 77. More references should be added from line 68 to line 77.
13- More previous studies about spectral indices should be added in introduction?
14- Please add the hint about the basic of remote sensing for detect physiological parameters? For example which area of spectrum regions should be used to detect them and what is basic of that?
15- From line 100 to 112. I think this section has no meaning. In the first, the authors give information about the chlorophyll which is related to nitrogen.
Please remove this section?
16- Please highlight in introduction, what is the novelty (originality) of the work? And what is new in your work that makes a difference in the body of knowledge?
17- I think, it is better to add Figure 1 in Materials and methods.
Materials and Methods
18- I do not know, what is the reason to add NDVI maps in figures 2 and 3. It came later.
19- Location in figure 2 should be presented in more details include longitude and altitude of this area of study.
20- Line 137. Please correct 9kg/666m2 to 9 kg/ 666 m2 and in line 141, line 150 and line 151. Please check the all manuscript?
21- Line 160. Please change this article to this work?
22- Line 164. Please change chlorophyll content to relative chlorophyll content or chlorophyll meter. SPAD could not measure chlorophyll content. You can say in whole manuscript chlorophyll meter or SPAD value.
23- Pleas adjust the table 2?
24- Please follow the journal format for writing the citations in table 3.
25- Please remove the equations from 1 to 4. It was presented in many papers. Write only the references of them
Results
26- Figure 4 is not clear to present the data of Rice biochemical variation. It is better to present this data in table. As well as support the text in the manuscript by numerical data. The explanation is more description.
27. Line 267. Please change ENVI to EVI.
28. Please add the significant levels of R2 in table 5?
29. At section 3.3. Single-factor regression modelling, please support the text by numerical data. The explanation is more description.
30. Regression modeling of rice physiological parameters based all vegetation indices or multi-factor should be presented. Please present the data in tables. Please combine all spectral indices in different models.
Discussions
31- The author presented the discussion in good way. But authors still need to support the discussions section according to comment number 30.
32- Please, write the practical applications of your work in a separate section, before the conclusions and provide your good perspectives.
Conclusion
32- Please write about the limitations of this work in details in conclusion section.
Author Response
Journal Name: Remote Sensing
Manuscript ID: remotesensing-2086518
Manuscript Title: Quantification of physiological parameters of rice varieties based on multi-spectral remote sensing and Machine Learning models
Dear Editor,
Thank you for your letter and careful consideration of our manuscript remotesensing-2086518 entitled “Quantification of physiological parameters of rice varieties based on multi-spectral remote sensing and Machine Learning models”. We appreciate the opportunity to revise and resubmit our manuscript. The comments were extremely valuable and have helped us to improve our manuscript greatly. We have modified the manuscript accordingly and hoped the revised manuscript would meet your requests. And then detailed corrections are listed below point by point.
Comments :
Point 1: Title: please added in the title (the machine learning models) the title should be (Quantification physiological parameters of rice varieties based on multi-spectral remote sensing and Machine Learning models). It better to reflect the all information about the manuscript in the title.
Responses:
ü Thank you for your comments. The title have been modified according to the comments.
Point 2: Line 21 to line 23. The sentence (For plant physiological indicators from multispectral observations….. …. based on the machine learning) from line 21 to 23 should be rephrased.
Responses:
ü Thank you for your comments. After carefully reading these articles, we have modified the original sentence.
Point 3: Line 24. Please do not use pronouns in the formulation of the sentence. For example, we quantify.
Responses:
ü Thank you for your comments. I has revised the article according to the comments to avoid using pronouns in the article.
Point 4: Line 24 to line 25. This sentence is wrong (we quantify the effect of reflectance spectra on the relationship between the plant physiological indicators such as the chlorophyll content, hole leaf area and dry matter accumulation). Spectral reflections are affected by the change in physiological characteristics but not affected in physiological characteristics. Please remove this sentence or rephrasing it.
Responses:
ü Thank you for your comments. Here is the author's mistake of meaning expression, and the article opinion is changed to “Spectral reflections are affected by the change in physiological characteristics”.
Point 5: Line 25. Do you mean hole or whole?
Responses:
ü Thank you for your comments. The method of measuring hole leaf area proposed in this paper is different of ordinary leaf area. The method of measuring the hole leaf area is to select 5 representative rice holes and determine the average leaf area of each rice hole to represent the mean value of the plot. The previous studies showed that the hole leaf area can well reflect the growth of rice, which is an important physiological parameter.
Point 6: Line 26. Please write the full name of the spectral indices (EVI, GNDVI, LCI, NDVI, NLI, OSAVI, RVI, SAVI, SIPI2) or remove them.
Responses:
ü Thank you for your comments. I have deleted the spectral indices.
Point 7: The best results of spectral indices should be added in abstract.
Responses:
ü Thank you for your comments. The article has added the best result, “Using the SVM model to predict the SPAD value of the plant, the Mean Squared Error (MSE), Root Mean Squared Error (RMSE), Mean Absolute Error (MAE), Mean Absolute Percentage Error (MAPE) and Symmetric Mean Absolute Percentage Error (SMAPE) value of the model were 1.90, 1.38, 0.13, 0.86 and 4.13, respectively”.
Point 8: The best results of SVM should be added in abstract.
Responses:
ü Thank you for your comments. The article has added the best result, “Using the SVM model to predict the SPAD value of the plant, the Mean Squared Error (MSE), Root Mean Squared Error (RMSE), Mean Absolute Error (MAE), Mean Absolute Percentage Error (MAPE) and Symmetric Mean Absolute Percentage Error (SMAPE) value of the model were 1.90, 1.38, 0.13, 0.86 and 4.13, respectively”.
Point 9: Please add the references for the first sentence in introduction?
Responses:
ü Thank you for your comments. I have added the reference according to the comments of the reviewer, The original text is modified as “Rice serves as the staple food for half of the world’s population [1]”
Point 10: Line 57. Change [8,9] to [8,9].
Responses:
ü Thank you for your comments. It has been modified according to the comments.
Point 11: Line 60 to line 63. References should be added from line 60 to line 63.
Responses:
ü Thank you for your comments. It has been modified according to the comments.
Point 12: Line 68 to line 77. More references should be added from line 68 to line 77.
Responses:
ü Thank you for your comments. It has been modified according to the comments.
Point 13: More previous studies about spectral indices should be added in introduction?
Responses:
ü Thank you for your comments. I have added other studies to the introduction part.
Point 14: Please add the hint about the basic of remote sensing for detect physiological parameters? For example, which area of spectrum regions should be used to detect them and what is basic of that?
Responses:
ü Thank you for your comments. It has been modified according to the comments of the reviewer.
Point 15: From line 100 to 112. I think this section has no meaning. In the first, the authors give information about the chlorophyll which is related to nitrogen. Please remove this section?
Responses:
ü Thank you for your comments. After consideration, this part does not really have much to do with the main idea of the article, and has been deleted.
Point 16: Please highlight in introduction, what is the novelty (originality) of the work? And what is new in your work that makes a difference in the body of knowledge?
Responses:
ü Thank you for your comments. The creativity of this article is mainly to use multiple vegetation indexes to jointly predict the physiological parameters. According to the comments of the reviewers, the following contents are added to the last paragraph.
Point 17: I think, it is better to add Figure 1 in Materials and methods.
Responses:
ü Thank you for your comments. It has been revised according to the comments of the reviewer, The picture has been moved to Materials and methods.
Point 18: I do not know, what is the reason to add NDVI maps in figures 2 and 3. It came later.
Responses:
ü Thank you for your comments. After checked the articles, we modified Figure 2 and replaced the NDVI photos with visible light photos. But the author thinks that Figure 3 shows the design of experimental plot, and using NDVI pictures can better show the difference of plant growth under different fertilization conditions.
Point 19: Location in figure 2 should be presented in more details include longitude and altitude of this area of study.
Responses:
ü Thank you for your comments. Figure 2 has been modified as required by the reviewer.
Point 20: Line 137. Please correct 9kg/666m2 to 9 kg/ 666 m2 and in line 141, line 150 and line 151. Please check the all manuscript?
Responses:
ü Thank you for your comments. The article has been revised as required by the reviewer.
Point 21: Line 160. Please change this article to this work?
Responses:
ü Thank you for your comments. The article has been revised as required by the reviewer.
Point 22: Line 164. Please change chlorophyll content to relative chlorophyll content or chlorophyll meter. SPAD could not measure chlorophyll content. You can say in whole manuscript chlorophyll meter or SPAD value.
Responses:
ü Thank you for your comments. It has been revised according to the comments of the reviewer, and the full text is revised to SPAD value. At the same time, the relationship between SPAD value and chlorophyll value is described in the foreword.
Point 23: Pleas adjust the table 2?
Responses:
ü Thank you for your comments. The article has been revised as required by the reviewer.
Point 24: Please follow the journal format for writing the citations in table 3.
Responses:
ü Thank you for your comments. The format of the form has been adjusted according to the comments of the reviewer, making the form more concise.
Point 25: Please remove the equations from 1 to 4. It was presented in many papers. Write only the references of them
Responses:
ü Thank you for your comments. I have deleted the equations from 1 to 4.
Point 26: Figure 4 is not clear to present the data of Rice biochemical variation. It is better to present this data in table. As well as support the text in the manuscript by numerical data. The explanation is more description.
Responses:
ü Thank you for your comments. The article provides detailed data of rice biochemical variation in the appendix, which is indicated at the end of the paragraph.
Point 27: Line 267. Please change ENVI to EVI.
Responses:
ü Thank you for your comments. The article has been revised as required by the reviewer.
Point 28: Please add the significant levels of R2 in table 5?
Responses:
ü Thank you for your comments. The article was revised according to the comments of the reviewer, and relevant instructions were added to the article.
Point 29: At section 3.3. Single-factor regression modelling, please support the text by numerical data. The explanation is more description.
Responses:
ü Thank you for your comments. According to the comments of the reviewers, Appendix 2 is added to the article to detail other indicators of the single factor regression model.
Point 30: Regression modeling of rice physiological parameters based all vegetation indices or multi-factor should be presented. Please present the data in tables. Please combine all spectral indices in different models.
Responses:
ü Thank you for your comments. Considering the powerful data analysis ability of machine learning, and providing more abundant spectral data for the prediction of physiological indexes, this paper uses all the nine spectral indexes mentioned as inputs to build a regression model. At the same time, the author also revised the article, indicating that all 9 vegetation indexes should be used.
Point 31: The author presented the discussion in good way. But authors still need to support the discussions section according to comment number 30.
Responses:
ü Thank you for your comments. Combined with previous studies, we add the discussion of multi-factor model and single factor model. The results show that the effect of using multi vegetation index to predict chlorophyll concentration is significantly better than using single vegetation index, which is of significance to continue the research. Our results were similar to the previous research and have added to the discussion part.
Point 32: Please, write the practical applications of your work in a separate section, before the conclusions and provide your good perspectives.
Responses:
ü Thank you for your comments. After carefully reading these articles, we add practical applications of my work in a separate section. The focus of our study is monitoring rice physiology. This method is not only helpful for monitor crop growth in the field, but also for several other applications worth mentioning, including quickly collect plant traits and screen excellent varieties.
Point 33: Please write about the limitations of this work in details in conclusion section.
Responses:
ü Thank for your comments. After carefully reading these articles, we are sorted out the limitations of the study. First, the model does not consider the image of the model caused by environmental factors such as weather. Second, the input of this model can only be high-precision multispectral images, which is difficult to apply to other remote sensing data. Third, since the data are collected in Guangdong, the model is only applicable to Guangdong.
This revised manuscript has not been published or presented elsewhere in part or in entirety, and is not under consideration by another journal. All the authors have approved the manuscript and agree with submission to your esteemed journal. There are no conflicts of interest to declare. Thank you for your consideration. I look forward to hearing from you.
Sincerely,
Lei Zhang
College of Agriculture, South China Agricultural University,
Guangzhou, 510642, PR China
Phone: (86)-20-85280203
Fax: (86)-20-85280203
E-mail: zhanglei@scau.edu.cn

Reviewer 4 Report
(1) In the abstract: it can also be extended to the detection and inversion of other key variables of crops. What’s specific variables?
(2) It is inappropriate for the technical route to appear in the introduction. It should be described in materials and methods.
(3) In the analysis of single-factor regression model accuracy, more precision evaluation indicators should be considered, not only the Mean Absolute Percentage Error.
(4) In terms of correct style of chapter headings/ references/ tables/ figure /units /decimals which are occasionally wrong formatted the authors can consult: journal recommendations for authors.
(5) There are some type of typos in your MS. Please double check you text and correct them all. Especially, the spelling and grammar problems are very obvious, it is suggested that native English speakers be approached for revision.
(6) The paper lists many related research algorithms, but it needs to be classified and summarized according to the characteristics of the method, and explain why you chose these methods.
Author Response
Journal Name: Remote Sensing
Manuscript ID: remotesensing-2086518
Manuscript Title: Quantification of physiological parameters of rice varieties based on multi-spectral remote sensing and Machine Learning models
Dear Editor,
Thank you for your letter and careful consideration of our manuscript remotesensing-2086518 entitled “Quantification of physiological parameters of rice varieties based on multi-spectral remote sensing and Machine Learning models”. We appreciate the opportunity to revise and resubmit our manuscript. The comments were extremely valuable and have helped us to improve our manuscript greatly. We have modified the manuscript accordingly and hoped the revised manuscript would meet your requests. And then detailed corrections are listed below point by point.
Comments :
Point 1: In the abstract: it can also be extended to the detection and inversion of other key variables of crops. What’s specific variables?
Responses:
ü Thank you for your comments. It can be transferred to other crops, such as measuring the chlorophyll content of corn/peanut and other crops by the same method, because the basic measurement principle is the same.
Point 2: It is inappropriate for the technical route to appear in the introduction. It should be described in materials and methods.
Responses:
ü Thank you for your comments. Modify according to the comments of the reviewer and move the picture to materials and methods.
Point 3: In the analysis of single-factor regression model accuracy, more precision evaluation indicators should be considered, not only the Mean Absolute Percentage Error.
Responses:
ü Thank you for your comments. After checked the single-factor regression model precision evaluation indicators data during our experiment, we have added other indicators, such as MSE, RMSE, SMAPE, MAPE and MAED. The added contents are listed in Appendix 2 in table form.
Point4: In terms of correct style of chapter headings/ references/ tables/ figure /units /decimals which are occasionally wrong formatted the authors can consult: journal recommendations for authors.
Responses:
ü Thank you for your comments. The article is modified according to journal recommendations by MDPI.
Point 5: There are some type of typos in your MS. Please double check you text and correct them all. Especially, the spelling and grammar problems are very obvious, it is suggested that native English speakers be approached for revision.
Responses:
ü Thank you for your comments. The manuscript has undergone English language editing by MDPI. The text has been checked for correct use of grammar and common technical terms, and edited to a level suitable for reporting research in a scholarly journal.
Point 6: The paper lists many related research algorithms, but it needs to be classified and summarized according to the characteristics of the method, and explain why you chose these methods.
Response:
ü Thank you for your opinion. Combined with the previous research, we modified the preface of the paper and added the discussion of different algorithms. In this part, we first compare machine learning with traditional algorithms, highlighting the advantages of machine learning: machine learning can rely on the powerful performance of modern computers, constantly reorganize knowledge structure, and achieve multiple iterations of self-improvement. Then we highlight previous studies on RF and SVM to illustrate the success of RF and SVM in processing remote sensing images.
This revised manuscript has not been published or presented elsewhere in part or in entirety, and is not under consideration by another journal. All the authors have approved the manuscript and agree with submission to your esteemed journal. There are no conflicts of interest to declare. Thank you for your consideration. I look forward to hearing from you.
Sincerely,
Lei Zhang
College of Agriculture, South China Agricultural University,
Guangzhou, 510642, PR China
Phone: (86)-20-85280203
Fax: (86)-20-85280203
E-mail: zhanglei@scau.edu.cn

Round 2
Reviewer 1 Report
At this stage, we can accept this manuscript because the authors prepared responses as per my comments.
Reviewer 3 Report
The authors improved the manuscript according to my comments and it could be accepted for publication.